# Noninvasive urinary protein signatures associated with colorectal cancer diagnosis and metastasis

Yulin Sun [1,9], Zhengguang Guo[2,9], Xiaoyan Liu[2], Lijun Yang[1], Zongpan Jing[1], Meng Cai[1], Zhaoxu Zheng[3], Chen Shao[4], Yefan Zhang[5], Haidan Sun[2], Li Wang[6], Minjie Wang[6], Jun Li[1], Lusong Tian[1], Yue Han[7], Shuangmei Zou[8], Jiajia Gao[1], Yan Zhao[1], Peng Nan[1], Xiufeng Xie[1], Fang Liu[1], Lanping Zhou[1], Wei Sun[2,10 ✉] & Xiaohang Zhao [1,10 ✉]

Currently, imaging, fecal immunochemical tests (FITs) and serum carcinoembryonic antigen (CEA) tests are not adequate for the early detection and evaluation of metastasis and recurrence in colorectal cancer (CRC). To comprehensively identify and validate more accurate noninvasive biomarkers in urine, we implement a staged discovery-verification-validation pipeline in 657 urine and 993 tissue samples from healthy controls and CRC patients with a distinct metastatic risk. The generated diagnostic signature combined with the FIT test reveals a significantly increased sensitivity (+21.2% in the training set, +43.7% in the validation set) compared to FIT alone. Moreover, the generated metastatic signature for risk stratification correctly predicts over 50% of CEA-negative metastatic patients. The tissue validation shows that elevated urinary protein biomarkers reflect their alterations in tissue. Here, we show promising urinary protein signatures and provide potential interventional targets to reliably detect CRC, although further multi-center external validation is needed to generalize the findings.

[1] State Key Laboratory of Molecular Oncology, National Cancer Center/National Clinical Research Center for Cancer/Cancer Hospital, Chinese Academy of Medical Sciences and Peking Union Medical College, Beijing, China. [2] Proteomics Research Center, Core Facility of Instruments, Institute of Basic Medical Sciences, Chinese Academy of Medical Sciences and Peking Union Medical College, Beijing, China. [3] Department of Colorectal Surgery, National Cancer Center/National Clinical Research Center for Cancer/Cancer Hospital, Chinese Academy of Medical Sciences and Peking Union Medical College, Beijing, China. [4] State Key Laboratory of Proteomics, Beijing Proteome Research Center, National Center for Protein Sciences (Beijing), Beijing Institute of Lifeomics, Beijing, China. [5] Department of Hepatobiliary Surgery, National Cancer Center/National Clinical Research Center for Cancer/Cancer Hospital, Chinese Academy of Medical Science and Peking Union Medical College, Beijing, China. [6] Department of Clinical Laboratory, National Cancer Center/ National Clinical Research Center for Cancer/Cancer Hospital, Chinese Academy of Medical Sciences and Peking Union Medical College, Beijing, China. [7] Department of Interventional Therapies, National Cancer Center/National Clinical Research Center for Cancer/Cancer Hospital, Chinese Academy of Medical Sciences and Peking Union Medical College, Beijing, China. [8] Department of Pathology, National Cancer Center/National Clinical Research Center for Cancer/Cancer Hospital, Chinese Academy of Medical Sciences and Peking Union Medical College, Beijing, China. [9] These authors contributed equally: Yulin Sun, Zhengguang Guo. [10] These authors jointly supervised this work: Wei Sun, Xiaohang Zhao. ✉email: sunwei@ibms.pumc.edu.cn; zhaoxh@cicams.ac.cn

Colorectal cancer (CRC) is the third most common malignancy and the second leading cause of cancer death globally, accounting for ~1 in 10 cancer cases and deaths[1]. It is estimated that the global burden of CRC will increase by 60% to >2.2 million new cases and 1.1 million cancer deaths by 2030[2].

Notably, the clinical stage at diagnosis, which is mainly defined by regional lymph node metastasis (LNM) and distant metastasis (DM), is the prognostic factor most directly related to the survival and recurrence of patients with CRC. For example, the 5-year relative survival of CRC patients was 90% for patients with stage I and II disease without metastasis, 71% for patients with LNM (stage III), and 14% for patients with distant spread (stage IV)[3]. However, ~35% and 20% of patients with newly diagnosed CRC present with LNM and synchronous distant metastases that were detected at or within 6 months of the initial diagnosis, respectively[3]. Moreover, ~25% of patients will develop delayed or metachronous distant liver metastases. Therefore, metastasis, including distant and high-risk lymph node spread, is the most important prognostic factor for survival in patients with CRC.

In the clinic, imaging and serum carcinoembryonic antigen (CEA) testing play central roles in monitoring the recurrence and metastasis of CRC. Computed tomography (CT) has a sensitivity and specificity of 51% and 85% for regional lymph nodes, 62% and 92% for distant lymph nodes, and ~71–73.5% and 96% for liver metastasis detection[4,5]. Magnetic resonance imaging (MRI) is superior to CT and has a sensitivity of 39–95% for lymph node metastases and 91–97% for liver metastases[5]. However, these mainstream imaging modalities also have some pitfalls, including high cost, poor detection of small lymph nodes or lesions (<1 cm), and they are unsuitable for patients with implants or impaired renal function[5]. In contrast, the sensitivity of serum CEA for patients with stage I, II, III, IV disease and recurrence is 4–11%, 25–30%, 38–44%, 65%, and 50–71%, respectively[6], with an overall specificity of ~70%[7]. Therefore, the current single surveillance strategy is not sufficient to evaluate metastasis and recurrence. There remains an urgent need for more accurate and noninvasive biomarkers in the clinic.

Urine is a source for discovering early and sensitive biomarkers because it can rapidly reflect changes in the body[8]. Moreover, its protein composition is significantly less complex than that of serum or plasma; thus, urine is a good sample for biomarker analyses[8]. Our study and other previous studies identified >8000 proteins in human urine[9]. Approximately 40% of urinary proteins originate from plasma proteins, and over 1800 proteins that are highly expressed in the colon can be detected in the urine of healthy individuals[9]. Therefore, it is feasible to identify noninvasive biomarkers for CRC in urine.

In previous studies, fragments of fibrinogen, hepcidin-20, and β2-microglobulin were found to be discriminative in urine samples between 76 cancer patients and 72 noncancer patients using SELDI-TOF and MALDI-TOF methods[10]. Urinary cysteine-rich protein 61 and trefoil factor 3 could yield a diagnostic capacity with an area under the receiver operating characteristic (ROC) curve (AUC) of 0.75 using ELISA in 176 CRC patients and healthy controls (HCs)[11]. A series of studies discovered and validated naturally occurring peptides (NOPs) in urine to discriminate CRC liver metastases from HCs using an LC-MS/MS method[12]. Hydroxylated collagen peptide (AGP) and two additional NOPs derived from collagen alpha-1(I) and collagen alpha-1(III), which were measured by multiple reaction monitoring and parallel reaction monitoring (PRM) approaches, respectively, were found to complement serum CEA to improve the detection of CRC liver metastases (15–20% increase in sensitivity)[13,14]. A previous study provided useful information on urinary biomarkers for detecting CRC. However, a comprehensive study, including discovery, validation, and verification in a large-scale cohort, is still unavailable until now.

In this study, we adopt a staged pipeline to develop urinary protein signatures of CRC for diagnosis and metastatic risk stratification in large-scale cohorts using urine and tumor tissue samples (Fig. 1). The performance of the signatures is evaluated and compared with that of FIT or serum CEA. Finally, the expression of key urinary proteins is validated in tissue specimens. This stepwise study yields highly accurate noninvasive urinary protein signatures and will improve the application of urinary proteomics in future CRC research.

## Results

**Clinical characteristics of urine specimens.** A total of 657 subjects, including HCs and CRC patients without metastases (NM), with LNM, and with DM, were recruited for this study. After excluding 105 individuals (Fig. 1), 552 qualified individuals were included for subsequent analyses. The detailed clinical information is shown in Table 1. The age and sex distributions were balanced among the HCs and three groups of CRC patients, except that the samples of the HC group in the dot blot analysis were a little younger. In addition, there were no statistically significant differences in the clinical parameters of histological differentiation grade, CA19-9 level, and tumor location among the three CRC groups (NM, LMN, and DM) in the TMT and PRM analysis; CEA level, CA19-9 level, and tumor location showed significant difference among the three CRC groups in the dot blot analysis.

**Discovery of differential urinary proteins using TMT approach.** First, CRC urinary protein candidates were discovered by the TMT labeling-2D-LC-MS/MS approach based on 36 subjects (Fig. 1). By using the criteria of a 1% false discovery rate (FDR) at both the peptide and protein levels, 2291, 2642, 3363, and 3181 proteins were identified in the four groups, respectively. The median technical and interindividual CVs were 7.3% and 22.2% in the four groups, respectively. By excluding the proteins with a technical CV > 30% and with an interindividual CV > 60% (exclusion of proteins with approximately top 5% interindividual CV) in each group, a total of 1976, 2151, 2634, and 2771 proteins were quantified in the four groups, respectively (Supplementary Data 1).

Unsupervised principal component analysis (PCA) of 995 common proteins in the four groups was performed to visualize the urinary protein profiling differences among the HCs and patients with different stages of CRC. The results suggested apparent discrimination between the HC and CRC groups (Fig. 2a). In the orthogonal partial least squares discriminant analysis (OPLS-DA) model, the four groups could be clearly separated (Supplementary Fig. 1a). One hundred permutation tests indicated no overfitting of the models (Supplementary Fig. 1b). Similar results were observed in the unsupervised clustering analysis of the average protein quantitation value for each group (Supplementary Fig. 1c).

The pairwise differential urinary proteins between NM, LNM, or DM, and HC were defined using a criterion of a mean fold change ≥1.5; thus, a total of 273, 337, and 355 proteins were identified, respectively (Supplementary Data 1). By ingenuity pathway analysis (IPA) of differential proteins, a pathway-pathway interaction diagram was generated by connecting all pairs of interacting pathways/diseases/functions. These differential proteins were enriched in tumor-related pathways, including tumor growth, tumor invasion, immune response, metabolism, and signaling (RAC, FAK, CDC42, and RhoA) pathways (Fig. 2b, Supplementary Data 2, 3).

Using the same criteria, the differential proteins within the three stages of CRC were defined in each pairwise comparison,

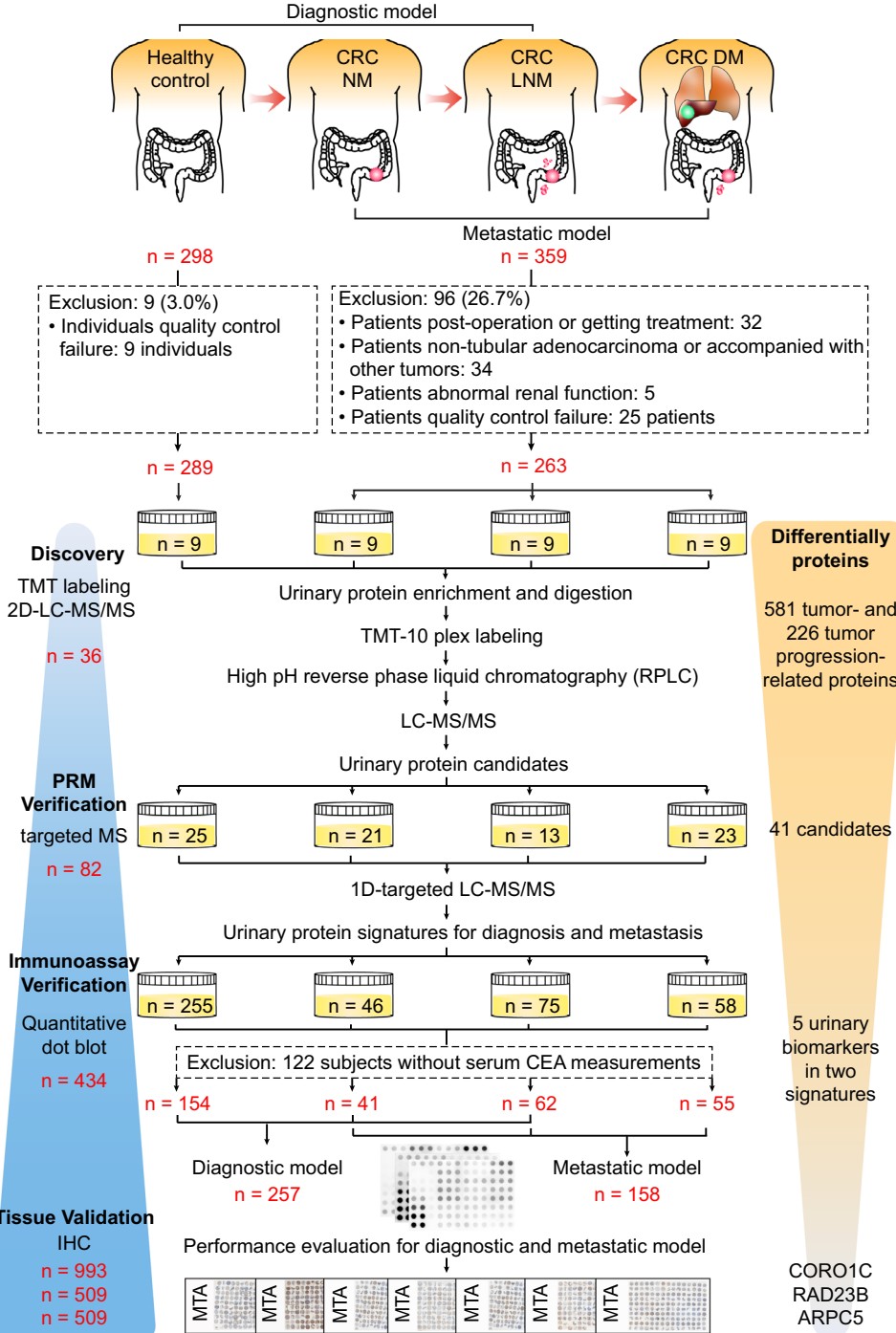

**Fig. 1 The overall workflow of study sample inclusion and exclusion criteria as well as the discovery, PRM verification, immunoassay verification, and tissue validation for CRC urine biomarkers.** The detailed inclusion and exclusion criteria of the samples are shown. CRC patients were divided into three groups by metastatic status: patients without metastases (NM), patients with lymph node metastasis (LNM), and patients with distant metastasis (DM). The four-stage workflow consisted of a series of mass spectrometry (MS) and immunoassay-based approaches, including the tandem mass tag (TMT) labeling-2D-LC-MS/MS quantitative proteomic strategy, parallel reaction monitoring (PRM)-based targeted proteomic method, quantitative dot blot analysis and tissue immunohistochemistry (IHC), to construct a coherent and high-throughput cancer biomarker method in urine. *CRC* colorectal cancer, *MTA* multi-tissue array.

and 93, 69, and 114 differential proteins were identified, respectively (Supplementary Data 1). IPA showed that tumor invasion-, immune response-, hemostasis-, angiogenesis-, and metabolism-related pathways/functions were enriched and tightly connected (Fig. 2c and Supplementary Data 4, 5).

Furthermore, in the IPA of disease and bio function analysis, cell death- and apoptosis-related proteins were increasingly inhibited, whereas the proteins in tumor proliferation, migration, and protein metabolism modules were gradually activated along with the development and progression of CRC. The immune response module was activated only in the early-stage of CRC (NM group), while tumor invasion-related proteins were activated only in the late stage of CRC (DM group) (Fig. 2d and Supplementary Data 6). Canonical pathway analysis revealed

**Table 1 Clinical information of all samples used in this study.**

| | TMT | | | | | PRM | | | | | Dot blot | | | | |
|---|---|---|---|---|---|---|---|---|---|---|---|---|---|---|---|
| | HC | NM | LNM | DM | P value | HC | NM | LNM | DM | P value | HC | NM | LNM | DM | P value |
| Sex | | | | | 0.6482a | | | | | 0.3792 | | | | | 0.1811 |
| Male | 4 | 6 | 3 | 4 | | 14 | 14 | 10 | 18 | | 153 | 34 | 51 | 40 | |
| Female | 5 | 3 | 6 | 5 | | 11 | 7 | 3 | 5 | | 102 | 12 | 24 | 18 | |
| Age (years) | | | | | 0.6354b | | | | | 0.3971b | | | | | 3.1E-05b |
| Median (min-max) | 48 (42–66) | 56 (35–71) | 57 (34–75) | 50 (40–80) | | 55 (40–68) | 56 (31–74) | 56 (45–80) | 58 (40–68) | | 56 (23–78) | 61 (26–78) | 62 (31–87) | 63 (39–87) | |
| Histological grade | | | | | 0.0513a | | | | | 0.9537a | | | | | 0.4372 |
| Well-differentiated | | 2 | 0 | 1 | | | 1 | 0 | 0 | | | 1 | 0 | 1 | |
| Moderately differentiated | | 7 | 5 | 5 | | | 16 | 9 | 16 | | | 30 | 40 | 31 | |
| Poorly differentiated | | 0 | 4 | 0 | | | 4 | 1 | 3 | | | 14 | 30 | 18 | |
| Unknown | | 0 | 0 | 3 | | | 0 | 3 | 4 | | | 1 | 5 | 8 | |
| CEA (ng/mL) | | | | | 0.0491a | | | | | 0.0044a | | | | | 1.3E-05c |
| <5 | - | 4 | 3 | 2 | | - | 6 | 4 | 5 | | 147 | 32 | 34 | 16 | |
| ≥5 | - | 1 | 0 | 6 | | - | 0 | 2 | 13 | | 7 | 9 | 32 | 39 | |
| Unknown | - | 4 | 6 | 1 | | - | 15 | 7 | 5 | | 101 | 5 | 9 | 3 | |
| CA19-9 (U/mL) | | | | | 0.0504a | | | | | 0.7809a | | | | | 0.00002c |
| <37 | - | 5 | 3 | 3 | | - | 5 | 6 | 11 | | 152 | 40 | 56 | 35 | |
| ≥37 | - | 0 | 0 | 5 | | - | 1 | 0 | 3 | | 2 | 1 | 7 | 20 | |
| Unknown | - | 4 | 6 | 1 | | - | 15 | 7 | 9 | | 101 | 5 | 12 | 3 | |
| Tumor location | | | | | 0.2977a | | | | | 0.1463a | | | | | 0.0083 |
| Right-sided | | 2 | 1 | 2 | | | 4 | 2 | 4 | | | 6 | 8 | 11 | |
| Left-sided | | 5 | 2 | 5 | | | 3 | 3 | 11 | | | 20 | 13 | 18 | |
| Rectum | | 2 | 6 | 2 | | | 14 | 8 | 8 | | | 20 | 54 | 29 | |
| TNM staging | | | | | 2.6E-11a | | | | | 2.0E-25a | | | | | 2.9E-82a |
| I | | 0 | 0 | 0 | | | 11 | 0 | 0 | | | 11 | 0 | 0 | |
| II | | 9 | 0 | 0 | | | 10 | 0 | 0 | | | 35 | 0 | 0 | |
| III | | 0 | 9 | 0 | | | 0 | 13 | 0 | | | 0 | 75 | 0 | |
| IV | | 0 | 0 | 9 | | | 0 | 0 | 23 | | | 0 | 0 | 58 | |

aTwo-sided P value for Fisher's Exact Test; bTwo-sided P value for one-way ANOVA or Kruskal–Wallis one-way analysis of variance; cStatistical analysis only among three CRC disease groups.

that tumor metabolism-related pathways and tumor survival-related pathways were activated in CRC. Tumor invasion-related pathways, such as the RAC, FAK, CDC42, and RhoA pathways, were increasingly activated along with the progression of CRC (Fig. 2e and Supplementary Data 7, and detailed pathway shown in Fig. 2f). Collectively, the above results indicated that urinary proteomics could reflect the enhanced tumor growth and malignancy status of CRC as well as the tumor invasion status in metastatic CRC.

**Verification of urinary proteins using PRM-based targeted MS**. We employed a widely used targeted high-throughput proteomics technology, PRM, to quantitatively verify the differential urinary proteins[15,16]. Among the CRC-related and CRC metastasis-related differential proteins identified in the discovery stage, 112 and 54 proteins showing gradient increasing or decreasing tendencies along with CRC progression, respectively, were selected for PRM verification. Among them, 77 proteins were identified in the PRM method design and then analyzed in 82 independent samples (Fig. 1). A pooled urine mixture was used as quality control (QC) to evaluate the system stability during the experimental process. Subsequently, 107 peptides from 66 proteins that passed the QC criteria were quantified. The average Pearson's correlation coefficient of QC samples was 0.99, indicating the repeatability of the QC samples and the stability of the MS platform (details in Methods).

After performing the PRM assay, 66 peptides from 41 differential proteins were successfully validated with trends consistent with the TMT approach in the four groups (Supplementary Data 8, 9). The heatmap and scatter plots of the 41 proteins (Fig. 3a) showed that 18 proteins were remarkably downregulated and 23 proteins were upregulated in CRC. Twelve upregulated proteins exhibited gradually increasing trends with CRC progression (Supplementary Data 9).

Multilevel analysis was used to define urinary protein signatures for CRC based on the 23 upregulated proteins. First, to obtain complementary biomarker combinations, the 23 proteins were evaluated by Spearman's rank correlation. Nine proteins with a moderately high correlation with more than five other proteins (rho ≥ 0.6) were excluded, and the 14 remaining proteins with less interdependency (median correlation coefficient of 0.34) were selected for subsequent analyses (Supplementary Fig. 2a).

Next, we evaluated these 14 proteins as input variables and identified the most important features in the diagnostic model and metastatic model using the random forest algorithm (Fig. 3b). Meanwhile, we measured the classification performance of each protein in the diagnosis and metastatic risk stratification using ROC analysis (Fig. 3c). The proteins with the top 10 highest values in the two models were chosen as candidate classifiers. Eight common proteins showed good performance for CRC diagnosis or metastatic risk stratification. Furthermore, the complementary performance of any two proteins was evaluated by comparing the combined AUC value for diagnosis and metastatic risk stratification (Fig. 3d). Finally, a urinary protein signature for CRC diagnosis consisted of CORO1C, ARPC5, and RAD23B, and a classifier for CRC metastasis consisted of CORO1C, RAD23B, GSPT2, and NDN.

The above diagnostic signature achieved 88.0% specificity, 75.8% sensitivity, and 81.0% accuracy with an AUC of 0.858 in the training set (Fig. 3e and Supplementary Table 1). Subsequently, 1000 bootstrap resamplings were applied to evaluate the extent of model overfitting. The bias-corrected AUC was 0.802, indicating the good robustness of the model.

To evaluate diagnostic signature performance in discriminating the NM, LNM, and DM groups from the HC group, three pairwise comparisons produced AUCs of 0.800, 0.948, and 0.935, respectively (Fig. 3e, Supplementary Fig. 2b and Supplementary Table 1), showing better prediction power for metastatic stages. For the patients in the NM group, the sensitivity, specificity, and accuracy were 65.0%, 88.0%, and 77.8%, respectively. Additionally, for early-stage I CRC patients, the above diagnostic signature achieved a sensitivity, specificity, accuracy, and AUC of 72.7%, 88.0%, 83.3%, and 0.782, respectively (Supplementary Fig. 2b).

The metastatic signature yielded a sensitivity, specificity, accuracy, and AUC of 81.1%, 70.0%, 77.2%, and 0.784, respectively, in discriminating between CRC patients with and without metastases (Fig. 3f and Supplementary Table 2). The bias-corrected AUC using 1000 bootstrap resamplings was 0.737, indicating the high robustness of the model. The performance of the metastatic signature in discriminating the three CRC groups was also analyzed in pairwise comparisons. The signature differentiated the NM group from the LNM and DM groups

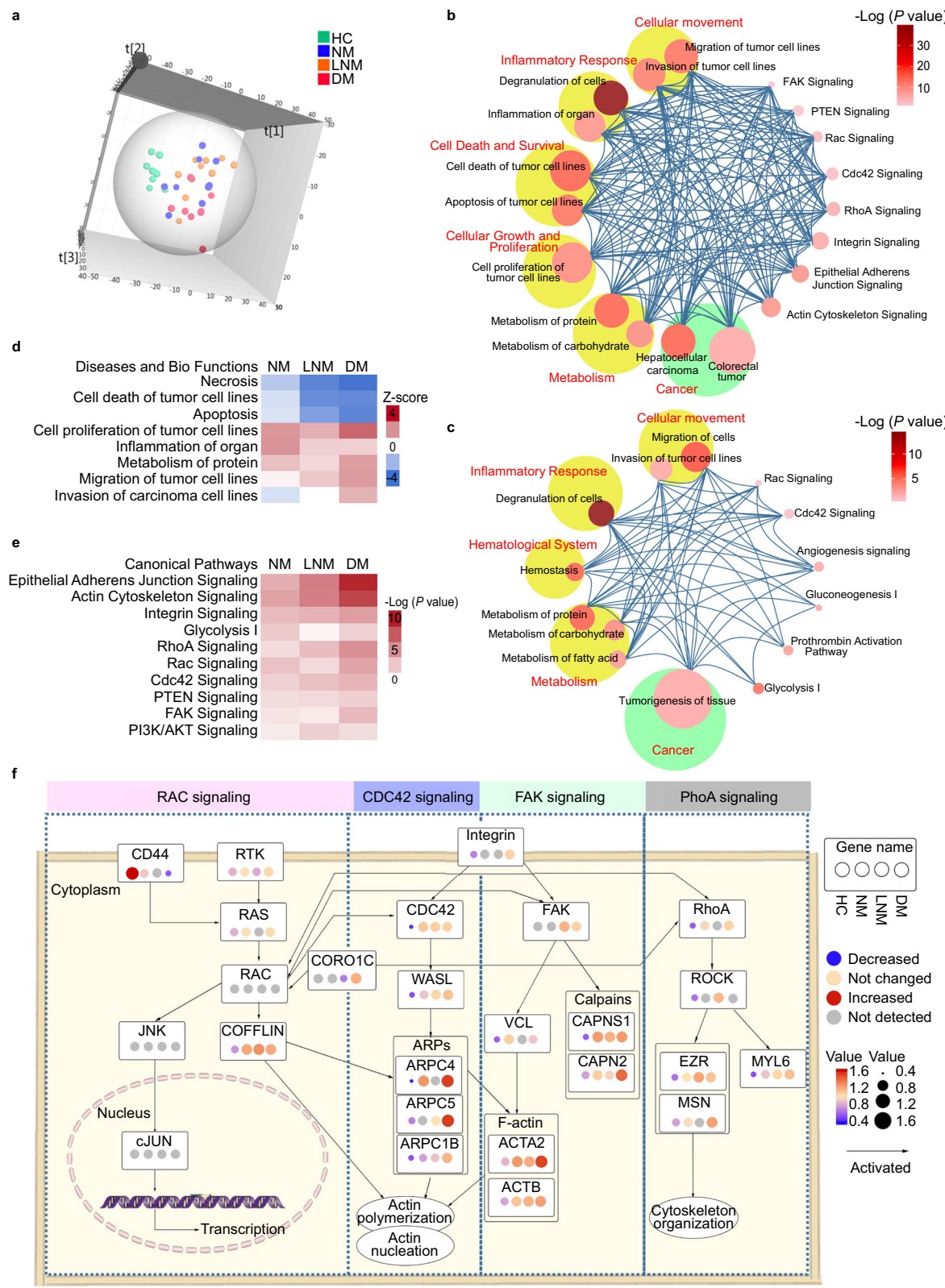

with sensitivity, specificity, accuracy, and AUCs of 69.2%, 70.0%, 69.7%, and 0.723 as well as 83.3%, 75.0%, 79.6%, and 0.827, respectively (Supplementary Table 2 and Supplementary Fig. 2c).

**Immunoassay verification of urinary protein signatures**. To validate urinary signatures to distinguish HCs and CRC patients

on a large-scale, we developed a quantitative dot blot detection system using urine[17] according to previous methods used with serum[17,18]. The urinary protein amount was quantified by standard curves of each protein and then calibrated by the corresponding urine creatinine measurement.

A total of 434 urine samples were recruited in the immunoassay verification stage (Fig. 1 and Supplementary

**Fig. 2 Quantitative urinary proteomics analysis in CRC at the discovery stage. a** Score plot of unsupervised principal component analysis (PCA) overview of urinary proteomics among the healthy controls (HCs), CRC without metastases (NM), CRC with lymph node metastasis (LNM) and CRC with distant metastasis (DM) groups. **b, c** CRC tumor-related (three CRC groups vs. HC; **b**) and tumor progression-related (**c**) pathway networks. Pathways are grouped vertically into three classes: disease, function, and canonical pathways. The color of each node represents the −log10 (P value) of that pathway. The size of each node represents the number of differential proteins in that pathway/disease/function. Interactions between pairs of pathways are indicated by curves. **d** Heatmap of the dysregulated biofunctions in the three CRC patient groups depicted by IPA. Red: Z_score>0, activated; Blue: Z score<0, inhibited. **e** Heatmap of the dysregulated canonical pathways in the three CRC groups depicted by IPA. The color represents the −log10 (P value) of that pathway. **f** Schematic diagram of tumor progression-related pathways, including the RAC, CDC42, FAK, and RhoA signaling pathways. The protein levels in the HC, NM, LNM, and DM groups are shown. The color and the size of the circle within each gene represent the expression levels of different stages of CRC for each gene. *CRC* colorectal cancer, *IPA* ingenuity pathway analysis. In **b**, **c**, **e**, the P value is calculated using the right-tailed Fisher's exact test without adjustments. The source data are provided in Source Data.

Data 10). The concentrations of CORO1C, ARPC5, RAD23B, GSPT2, and NDN were significantly higher in the urine of patients with CRC than in the urine of HCs (P < 0.0001 for all; Fig. 4a). In the three CRC groups, the levels of these five urinary proteins showed a gradient with an increasing trend that correlated with disease progression, achieving the highest levels in DM (P < 0.01 for all; Fig. 4a).

**Performance of urinary diagnostic and metastatic signatures.** In the 434 urine samples detected by immunoassay, serum CEA measurements were available in 312 samples, including samples from 154 HCs and 158 CRC patients. To facilitate the comparison of CEA results, these 312 samples were used to further analyze (Fig. 1). Because the PRM data were the peptide-level quantitatively intensities based on the mass spectrometry (MS) signals, whereas the antibody-based immunoassay produced the densitometric data at the protein level, the machine learning models trained on PRM platform cannot be directly applied to the immunoassay platform. Therefore, to validate the results of PRM analysis, the models of immunoassay were re-trained using the same protein signatures for CRC diagnosis and metastasis risk stratification.

For the diagnostic model (NM + LMN vs. HC), in the training set (68 NM and LNM vs. 103 HCs, n = 171), the signature (CORO1C, ARPC5, and RAD23B) had an AUC, sensitivity, specificity, and accuracy of 0.787, 69.1%, 79.6%, and 78.0%, respectively (Fig. 4b and Supplementary Table 3). Internal 10-fold cross-validation was performed, and the validated AUC, sensitivity, specificity, and accuracy were 0.777, 67.6%, 80.6%, and 77.9%, respectively. In the validation set (51 HCs vs. 35 CRC, total n = 86), the signature achieved an AUC, sensitivity, and specificity of 0.846, 74.3%, and 86.3%, respectively (Supplementary Fig. 3a and Supplementary Table 3). Moreover, the discriminative capacity of the diagnostic signature was strengthened in the NM, LNM, or DM group versus the HC group, yielding AUCs of 0.796, 0.814, and 0.913, respectively (Fig. 4b and Supplementary Fig. 3a). For the patients in the NM group, the sensitivity, specificity, and accuracy were 73.2%, 79.2%, and 78.0%, respectively. Additionally, for early-stage I CRC patients, the above diagnostic signature achieved a sensitivity, specificity, accuracy, and AUC of 90.9%, 83.3%, 82.8%, and 0.879, respectively (Supplementary Fig. 3a).

For the metastatic model, the signature consisting of CORO1C, RAD23B, GSPT2, and NDN was used to distinguish metastatic (62 LNM and 55 DM) and nonmetastatic (41 NM) CRC. Leave-one-out cross-validation method was used to evaluate the performance, yielding AUC, sensitivity, specificity, and accuracy of 0.699, 66.7%, 68.3%, and 67.1%, respectively (Fig. 4c). Moreover, the discriminative power of the metastatic signature was higher for DM versus NM (AUC 0.76) than for LNM versus NM (AUC 0.61) (Supplementary Fig. 3b and Supplementary

Table 4), highlighting its better performance for CRC with distant metastasis.

**Urinary protein signatures complemented FIT and serum CEA.** For comparison, the CRC screening biomarker FIT was also measured in 122 samples (training set: HC, n = 51; CRC, n = 33; validation set: HC n = 22; CRC, n = 16). The sensitivity and specificity of FIT were 66.7% (22/33) and 100% (51/51) in the training set and 50.0% (8/16) and 100% (22/22) in the validation set, respectively. Meanwhile, the urinary diagnostic signature achieved an AUC, sensitivity, and specificity of 0.812, 72.7%, and 86.3% in the training set and 0.864, 68.7%, and 95.5% in the validation set, respectively. The diagnostic sensitivity of the urinary signature was higher than that of FIT. In the training and validation cohorts, the urinary signature increased the diagnostic power in an additional 7 (21.2%) and 7 (43.7%) patients, respectively (Fig. 4d-upper). For FIT-negative patients, 63.6% (7/ 11) of patients in the training set and 87.5% (7/8) of patients in the validation set were correctly diagnosed by the diagnostic signature. The combination of FIT and the urinary diagnostic signature could achieve better diagnostic capability with a sensitivity and specificity of 81.8% and 94.1% in the training set and 93.8% and 86.4% in the validation set, respectively.

In the metastatic model, the clinical CRC metastasis biomarker CEA was measured in 158 serum samples. The overall performance of the urinary metastatic model was similar to serum CEA. And the combination of the urinary protein signature with serum CEA had a slightly better predictive power compared with CEA alone, with AUC, sensitivity, specificity, and accuracy values of 0.739, 70.9%, 73.2%, and 71.5%, respectively (Fig. 4c and Supplementary Table 4). Moreover, in the stratified discrimination of LNM and DM from NM, the combination yielded AUCs of 0.659 and 0.831, respectively (Supplementary Table 4 and Supplementary Fig. 3b). Furthermore, for the classification of metastatic and nonmetastatic CRC, at the most commonly used CEA threshold of 5 ng/mL in the clinic, the sensitivity, specificity, and accuracy of serum CEA were 58.1%, 78.0%, and 63.3%, respectively (Supplementary Table 4). In the patients with metastatic CRC, CEA was positive in 68 (58.1%) patients, and the urine metastatic signature increased the diagnostic power in an additional 29 (24.8%) patients. The subgroup analyses revealed that the discriminative power of the metastatic model increased in an additional 17 (27.4%) and 12 (21.8%) patients with LNM and DM, respectively, compared with that of CEA alone in 29 (46.8%) and 39 (70.9%) patients (Fig. 4d-bottom). Moreover, 51.5% (17 out of 33) or 75.0% (12 out of 16) of the CEA-negative LNM or DM patients were correctly predicted to have metastases using the panel.

To visualize the urinary protein performance on CRC diagnosis and metastatic risk stratification, with a specificity of 95%, the cutoff values of each protein and signature were used in the training and validation urine samples. In the diagnostic

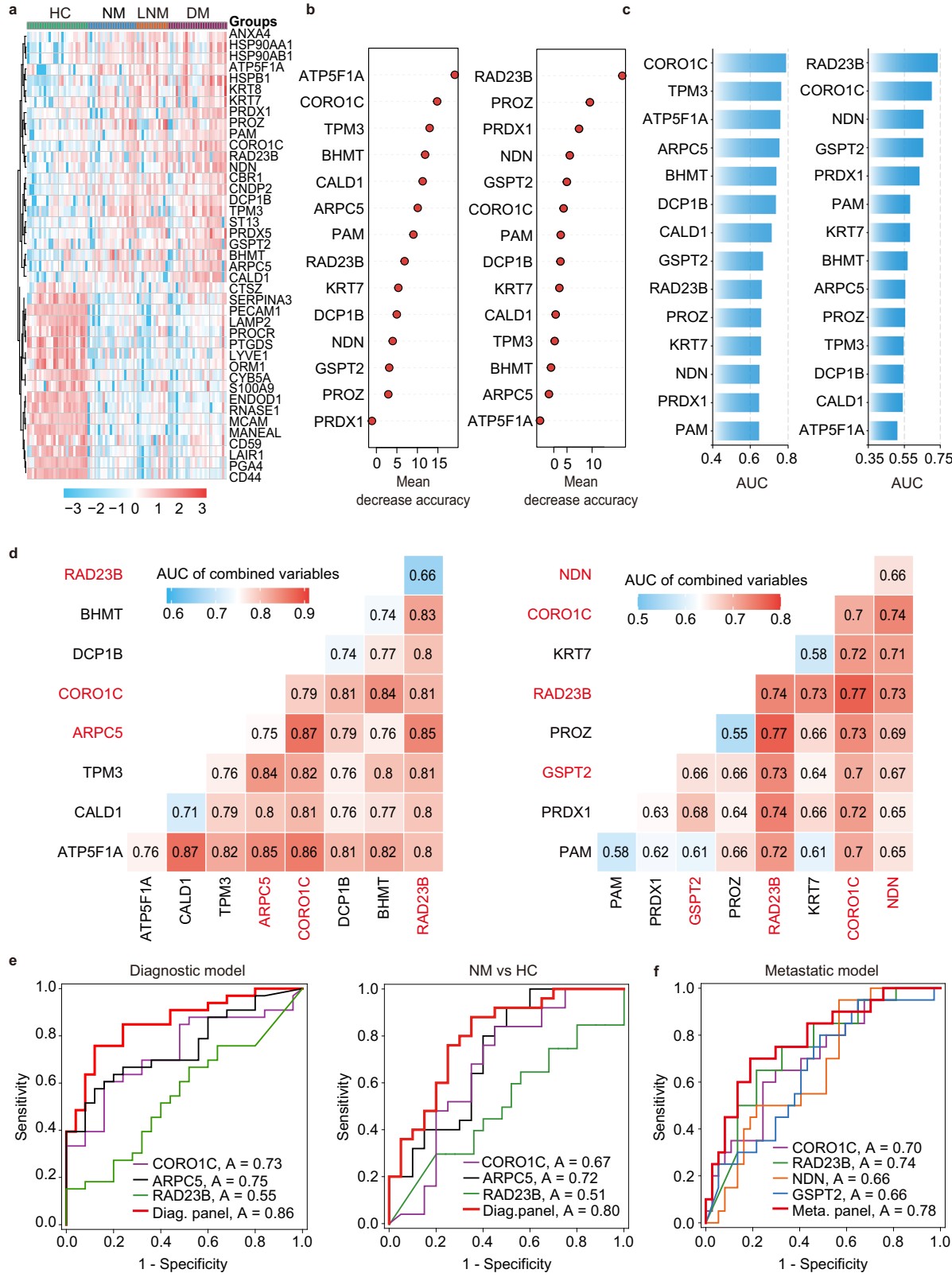

model, the combined signature with FIT led to a sensitivity and specificity of 81.8% and 94.1% in the training set and 93.8% and 86.4% in the validation set (Supplementary Table 5, Fig. 4e-upper, Supplementary Fig. 3c). In the metastatic model, the combined signature with serum CEA yielded a sensitivity and specificity of 71.8% and 75.6% (Supplementary Table 6 and Fig. 4e-bottom). The panel achieved higher sensitivity than the use of a single

protein. In addition, the panels in both models can complement FIT or serum CEA to provide better performance.

**Elevated tissue proteins reflect advanced stages and poor prognosis.** To evaluate whether the abnormal alterations of three diagnosis-related proteins in the urine originated from CRC

**Fig. 3 Generation of the CRC urinary protein biomarker signature. a** Unsupervised clustering analysis of 41 deregulated proteins in the four groups (HC, healthy controls; NM, CRC without metastases; LNM, CRC with lymph node metastasis; DM, CRC with distant metastasis) based on PRM data. **b** Variable importance plots produced by the random forest algorithm measured as each variable's mean decrease in accuracy. The most important predictors have the highest mean decrease accuracy values. Left panel, for the class of CRC patients vs. HCs (diagnostic model); right panel, for the class of patients with metastasis (LNM and DM) vs. NM (metastatic model). **c** The AUC was used to evaluate the ability of individual proteins to distinguish between CRC patients and HCs (left panel; diagnostic model) as well as between patients with metastasis and those without metastasis (right panel; metastatic model). **d** The AUC of combining any two variables was calculated and shown as matrix plots for the diagnostic model and metastatic model. The proteins that show superior discrimination and complementarity are marked in red. **e** ROC curves for the diagnostic model (NM + LNM vs. HC) to discriminate the HC group from the CRC group (NM + LNM) or NM group (stage I + stage II). **f** ROC curves for the metastatic model (LNM + DM vs. NM). The performance of the selected protein signature and individual proteins were compared. *CRC* colorectal cancer; *PRM* parallel reaction monitoring; *ROC*, receiver operating characteristic, *AUC* area under ROC curve, *Diag. panel*, diagnostic panel, *Meta. panel* metastatic panel. The source data are provided in Source Data.

tumor tissues, we performed an immunohistochemical staining assay in several tissue microarrays. A total of 961, 500, and 500 tumors as well as 836, 413, and 434 paracarcinoma normal tissues from 993, 509, and 509 subjects with colorectal adenocarcinoma were informative for CORO1C, RAD23B, and ARPC5, respectively (Fig. 1 and Supplementary Data 11–13).

As shown in Fig. 5a, CORO1C and ARPC5 were localized in the cytoplasm, whereas RAD23B was mainly localized in the cytoplasm and nucleus. In the adjacent noncancerous tissues, CORO1C, RAD23B, and ARPC5 staining remained relatively weak, with positive rates of 5.6% (47/836), 32.7% (135/413), and 9.2% (40/434), respectively. In contrast, positive immunostaining for CORO1C, RAD23B, and ARPC5 was observed in 55.7% (535/961), 84.0% (420/500), and 85.8% (429/500) of the CRC tumor tissues, respectively, showing significant upregulation compared with the adjacent noncancerous tissues (all $P < 0.0001$). We further divided the positive tumor staining cases into weak and strong expressions. For CORO1C, RAD23B, and ARPC5, weak and strong staining was found in 49.3% (264/535), 86.9% (365/420), and 39.9% (171/429) and 50.7% (271/535), 13.1% (55/420) and 60.1% (258/429) of cases, respectively. Subsequent clinical significance analysis showed that higher CORO1C levels were associated with depth of invasion (T staging, $P = 0.0035$) and distant metastasis (M staging, $P = 0.0210$; Fig. 5b). RAD23B was positively correlated with depth of invasion (T staging, $P = 0.0308$), LNM (N staging, $P = 0.0149$), distant metastasis (M staging, $P = 0.0116$) and TNM staging ($P = 0.0006$). ARPC5 levels were significantly increased in tumors invading the subserosa or visceral peritoneum of the colon or rectum ($P = 0.0021$) and advanced TNM staging ($P = 0.0495$).

The subsequent Kaplan–Meier survival analysis indicated that CRC patients with high CORO1C protein levels had shorter recurrence-free survival (RFS) times ($P = 0.0075$) but not overall survival times ($P = 0.9171$; Fig. 6). RAD23B was significantly correlated with unfavorable overall survival ($P = 0.0124$) but not RFS ($P = 0.1123$) in CRC patients (Fig. 6). ARPC5, however, had no prognostic significance in CRC (Fig. 6). Taken together, since these two urinary proteins were shared by both the diagnostic and metastatic signatures, high levels of CORO1C and RAD23B tissue expression indeed promoted the metastatic potential of malignant cells in CRC.

In addition, we also observed the expression of these three proteins in normal colon or rectum epithelium ($n = 8$), low-grade intraepithelial neoplasia (LGIN) lesions ($n = 21$), and high-grade intraepithelial neoplasia (HGIN) lesions ($n = 41$) (Supplementary Fig. 4). Intriguingly, CORO1C, RAD23B, and ARPC5 were barely visible or showed relatively weak expression in the normal mucosal epithelium. Moreover, the protein expression of CORO1C and RAD23B gradually increased with the progression of LGIN to HGIN. The level of ARPC5, however, was dramatically increased at the LGIN stage. Thus, the overexpression of three urinary proteins in the diagnostic signature may occur at the precancerous stage of CRC.

## Discussion

CRC is a highly heterogeneous disease. Recent genomic and transcriptomic analyses indicate that CRC metastasis, including LNM and distant metastasis, is partly mediated through a polyclonal seeding mechanism in at least one-third of patients[19,20]. Therefore, regional spread and distant metastases of CRC reflect the inherent characteristics of primary tumors, and both share a similar genetic basis but also show significant differences.

Since the acquisition of metastatic capacity is an early event in tumorigenesis, it provides a possibility for predicting regional or distant dissemination based on the analysis of primary tumor tissues or human body fluids. However, high heterogeneity of primary tumors may result in inaccurate results based on transcriptomic and proteomic profiling of primary tumors, especially for biopsy specimens. Thus, body fluids that reflect the general change in pathophysiological status may serve as good sources for finding biomarkers for metastasis risk stratification and early diagnosis.

In this study, we systematically analyzed urinary proteins for the diagnostic and prognostic prediction of CRC in combination with quantitative proteomics, targeted proteomics, and immunoassays. To date, this is the largest and most comprehensive study to determine noninvasive biomarkers of CRC. The specificity of our diagnostic and metastatic signatures for CRC, rather than in other tumors, needs further evaluation.

In the discovery stage, a series of CRC-related differential proteins were identified. Functional analysis showed that the urinary proteome could reflect the hallmark features of cancer, including sustaining cell proliferation, resisting cell death, reprogramming energy metabolism, etc.[21]. In addition, the urinary proteome also reflected the characteristics of CRC in different stages. For example, we found that the immune response module was activated only in early-stage CRC (NM group), which is in line with the results of a previous study at the tissue level[22]. Importantly, tumor invasion-related proteins and pathways represented by RAC, FAK, CDC42, and RhoA signaling were enriched only in the DM group, demonstrating unique characteristics of metastatic CRC.

Previous studies (Supplementary Data 14) identified 5 diagnostic and 2 metastatic proteins of CRC in urine[10–14]. In our study, using high-throughput TMT methods, a total of 581 diagnostic and 226 metastatic proteins were identified, of which 4 diagnostic markers were consistent with those identified in previous studies. In CRC serum/plasma proteomics studies, 54 diagnostics and 13 metastatic proteins were identified in both this study and previous works. Notably, by comparing the 31 cancer-associated proteins identified by the CPTAC CRC project[23] with the proteins found in our study, 3 proteins showed similar expression trends, including S100P, CTHRC1, and S100A11. Therefore, our urinary proteomic analysis reflected the changes in CRC tumor tissues.

At the PRM and immunoassay verification stages of this study, a panel of three urinary biomarkers (CORO1C, ARPC5, and

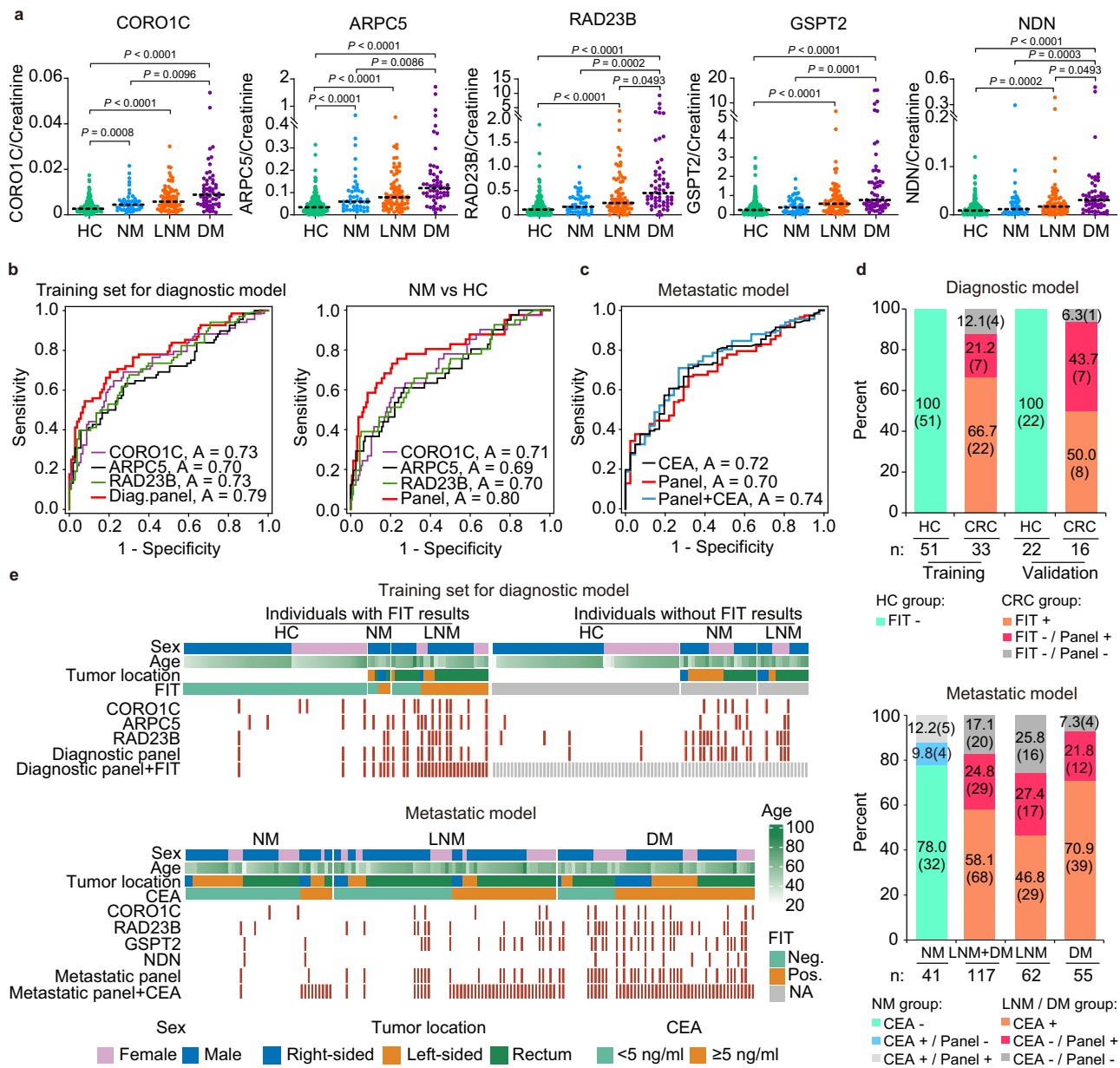

**Fig. 4 Independent urine verification of the urinary protein signature using dot blot analysis. a** Scatter plot for CORO1C, ARPC5, RAD23B, GSPT2, and NDN in 255 healthy controls (HCs) and 179 CRC patients, including CRC without metastases (NM; $n = 46$), CRC with lymph node metastasis (LNM; $n = 75$) and CRC with distant metastasis (DM; $n = 58$). The median values in each group are shown as black dotted lines. The differences between groups for each marker were analyzed by two-sided Kruskal–Wallis test followed by a Dunn's multiple comparisons test. Uncropped original blots in Source Data. **b** ROC curve of the diagnostic panel for the diagnostic model (NM + LNM vs. HC) in the training set and discrimination of the NM group CRC from HC. **c** ROC curve of serum CEA, metastatic panel, and the combination of the metastatic panel and CEA for the metastatic model (LNM + DM vs. NM). **d** Diagnostic and metastatic predictive power of the diagnostic signature and metastatic signature in the individuals who were misdiagnosed by the FIT test or serum CEA. The values in parentheses indicate the number of samples corresponding to each percent. +, positive; −, negative; $n$, number of samples. **e** Heatmap of the dot plot data for single urinary markers as well as the diagnostic or metastatic panel with a specificity of 95%, and the combination of corresponding clinical biomarker indices for the diagnostic or metastatic model was considered positive when either the panel or FIT/CEA was positive. Red: positive using the cutoff value with a specificity of 95%. The FIT test, serum CEA, tumor location, sex, and age are indicated by color-coding. *CRC* colorectal cancer, *FIT* fecal immunochemical test, *CEA* carcinoembryonic antigen, *Neg.* negative; *Pos.* positive; *NA* not available. The source data are provided in Source Data.

RAD23B) for CRC diagnosis and a panel of four urinary biomarkers (CORO1C, RAD23B, GSPT5, and NDN) for CRC metastatic risk stratification were defined. Among these proteins, CORO1C and ARPC5 are known regulators of actin cytoskeleton dynamics. CORO1C is necessary for the release of inactive RAC1 from the nonprotrusive membrane and the activation and redistribution of RAC1 to a protrusive tip; accordingly, the activation of RAC1 induces membrane ruffling and lamellipodia formation at the leading edge[24,25]. ARPC5 is one of the subunits of the Arp2/3 complex, which is the central actin nucleator that promotes branched filament formation and creates a complex cortical membrane actin network to generate the force necessary for protrusion[26]. RAD23B is involved in the nucleotide excision repair of damaged DNA, and its abnormal expression has been

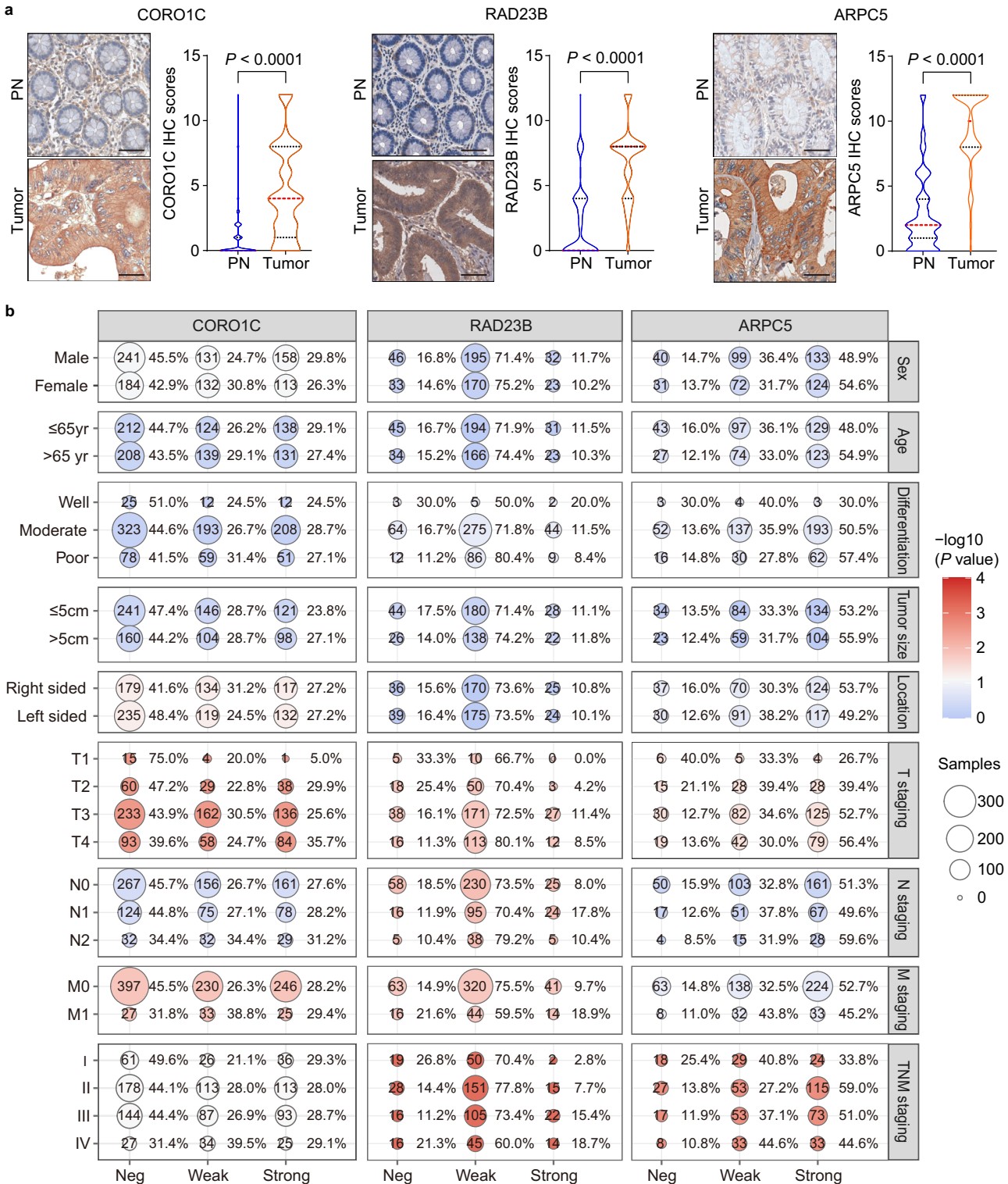

**Fig. 5 Immunohistochemical staining of three diagnostic biomarkers in tissues and their clinical significance. a** Representative immunohistochemistry images and score distribution of CORO1C, RAD23B, and ARPC5 expression in paracarcinoma normal tissues (PN) and CRC tumors (tumor). The median and quartile values in each group of individuals are shown as thick red dash lines and thin black dotted lines, respectively. Scale bar: 50 μm. The statistical analysis was performed by two-sided Mann–Whitney rank test. **b** The balloon plot for the clinical significance of CORO1C, RAD23B, and ARPC5 in colon adenocarcinoma patients with distinct staining intensities. The number in the circle is the sample size, and the percentage next to the circle is the corresponding percentage. Chi-square test was used for calculating the two-sided *P* values. *Neg* negative, *Weak* weak expression, *Str.* strong expression. The source data are provided in Source Data.

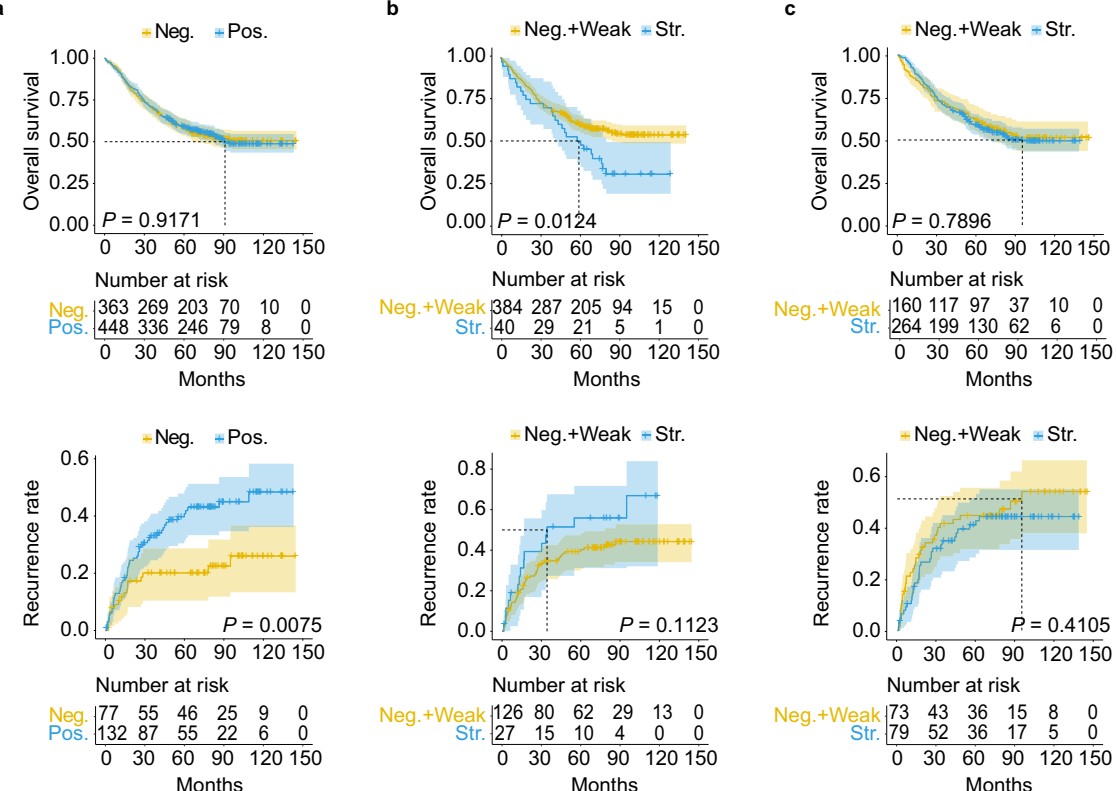

**Fig. 6 Kaplan–Meier survival analysis of three diagnostic biomarkers in patients with colorectal adenocarcinoma. a, b, c** The overall survival (OS) and recurrence-free survival (RFS) curves of CORO1C (**a**), RAD23B (**b**), and ARPC5 (**c**). The survival curve with 95% confidence interval in shading was plotted for each group. P values were calculated by two-sided log-rank test. The numbers at the bottom of each panel indicate the patients at risk. *Neg.* negative, *Pos.* positive, *Weak* weak expression, *Str.* strong expression. The source data are provided in Source Data.

found in breast cancer[27]. GSPT2 is a GTPase that mediates translation termination and has been reported to be a biomarker for hepatocellular carcinoma and CRC liver metastasis in serum[28,29]. NDN is a member of the melanoma-associated antigen family and serves as a candidate tumor suppressor gene to facilitate the entry of the cell into cell cycle arrest in multiple tumors, including CRC[30].

Currently, there are no relevant studies on the CORO1C, RAD23B, and ARPC5 proteins in CRC, except for our very recent work, which showed that RAD23B was overexpressed in CRC tumor tissues and associated with pathological grade, TNM staging, liver metastasis, and poor overall survival[31]. Mechanistically, RAD23B interacted with CORO1C to facilitate the aggregation of CORO1C and RAC1 to the lateral edges of CRC cells to form invasive protrusions and invadopodia, which enhanced the migratory and invasive abilities of CRC cells[31].

Notably, our urinary diagnostic signatures showed superior performance to the conventional FIT test used in the clinic. First, our diagnostic signature showed better sensitivity for patients with stage I and stage II CRC (NM group). In the training cohort, our diagnostic signature was positive in 66.7% of patients (23/33) versus 63.6% (21/33) for FIT, whereas in the validation set, they were 87.5% (14/16) versus 50.0% (8/16), respectively. Moreover, FIT greatly complemented our diagnostic signature. Overall, 63.6% and 87.5% of FIT-negative CRC patients in the training and validation sets, respectively, were correctly distinguished by the diagnostic signature. The combination of FIT with the diagnostic signature dramatically improved the sensitivity with a slight loss of specificity of FIT. It should be noted that the specificity of FIT was overestimated in this study because other upper gastrointestinal tumors and benign diseases were not

included in our samples. Thus, our urinary diagnostic signature is a potent biomarker panel for the detection of early- and intermediate-stage CRC with better accuracy than FIT alone.

Next, the main goal of the urinary metastatic model was to increase the sensitivity of CEA to identify LNM and distant metastasis more accurately in patients with a medical record of CRC. Compared with CEA alone, the combination of CEA (≥5 ng/mL) and the urinary metastatic signature increased the sensitivity from 58.1% to 82.9% (+24.8%) for all metastatic patients, from 46.8% to 74.2% (+27.4%) for LNM patients and from 70.9% to 92.7% (+21.8%) for DM patients. It needs to be noted that we use the leave-one-out cross-validation to build the metastatic model for risk stratification, since the small sample size of NM group does not allow to a split of the training and validation sets. Therefore, external validation still needs to be further evaluated in a larger cohort. And follow-up and prospective cohort studies are also needed to evaluate whether our urinary metastatic signature could provide a lead time for the detection of recurrent CRC.

To evaluate whether the alterations of urinary protein markers can reflect the changes in tissue levels, we performed tissue validation for the three diagnosis-related proteins using immunohistochemical staining. First, higher levels of CORO1C and RAD23B expression were positively correlated with tumor invasive depth, distant metastasis (M staging), and/or LNM (N staging). RAD23B and ARPC5 expression were also significantly correlated with TNM staging. Meanwhile, CORO1C and RAD23B were significantly associated with unfavorable recurrence-free survival and overall survival, respectively. In contrast, dot blot analysis in urine revealed that the abundance of all five urinary markers displayed a significant increase in DM

group compared with NM group (Fig. 4a). These results indicated that alterations in tissue levels may be well represented in urine.

Furthermore, we found that aberrantly high levels of CORO1C, RAD23B and ARPC5 expression in tissue begin to appear at precancerous lesions, especially in HGIN, hinting that they are sensitive to early detection. Combined with the findings that the CORO1C, RAD23B, and ARPC5 concentrations in urine show good performance in distinguishing early-stage CRC (stage 0 and stage I) from HCs, these findings strongly suggest alterations in urinary proteins may occur at the tumor initiation stage. Urinary proteomes have already been found to reflect early changes and predict tumor formation in various tumor-bearing animal models[32–34]. In human specimens, it was reported that urinary protein markers showed high diagnostic accuracy in identifying hepatocellular carcinoma among high-risk hepatitis C (HCV)-infected patients, stage I non-small cell lung cancer (NSCLC), and gastric cancer patients[35–37].

A recent study showed that in bladder cancer, CORO1C overexpression was positively associated with advanced disease, poor prognosis, EMT, and tumor-infiltrating neutrophils[38]. To evaluate whether the alteration of five markers in urine was related to the other urological tumors, we analyzed their urinary protein concentrations in patients with bladder cancer ($n = 20$) and renal cell carcinoma ($n = 22$) as well as their respective sex- and age-matched HCs ($n = 20$ and 20, respectively) using a PRM targeted proteomic strategy. The results showed that there was no significant difference between renal cell carcinoma or bladder cancer and HCs for the concentrations of these five proteins in urine (Supplementary Fig. 5 and Supplementary Data 15, 16), suggesting that these five urinary protein markers were relatively unique for CRC. The expression changes of these proteins in other tumors need further evaluation in the future.

In conclusion, this is currently the largest and most comprehensive urinary biomarker discovery study in CRC utilizing a discovery-verification-validation pipeline. The urinary diagnostic signature combined with FIT improved the sensitivity by 24.5% (85.7% vs. 61.2%) compared with FIT alone. Moreover, the metastatic signatures combined with serum CEA (≥5 ng/ml) improved the sensitivity by 24.8% (82.9% vs. 58.1%) for metastatic CRC patients compared with CEA alone. We also found that aberrantly high levels of diagnosis-related biomarkers are present in precancerous lesions and early-stage tumor tissues. The above results showed that the urinary proteome could comprehensively reflect the pathophysiological changes in different CRC stages, even in early-stage CRC. Our findings provide the promising urinary protein biomarkers to reliably diagnose and detect CRC, whether or not in combination with the FIT test, but also indicate potential interventional targets for metastatic CRC. Because only a small number of early-stage CRC patients were enrolled, the accuracy of the diagnosis model for detecting stage I CRC needs to be further validated. And external validation using larger cohorts, especially from multi-centers, is also needed to generalize the conclusions. Additionally, the underlying mechanism of these biomarkers during CRC progression is still unclear and will be explored in our future research.

## Methods

### Ethical issues
This study was approved by the Ethics Committee of the Institute of Basic Medical Sciences, Chinese Academy of Medical Sciences (#047-2019) with an exemption of informed consent and was performed according to the Declaration of Helsinki Principles.

### Experimental design
The objectives of the present study were to systemically identify and validate potential noninvasive diagnostic and metastatic predictive markers for CRC in urine. Therefore, three groups of CRC patients with distinct metastatic risk were recruited: CRC patients without any metastases, with LNM, and with distant metastasis. In addition, a four-stage workflow consisting of a series

of MS and immunoassay-based approaches, including a TMT labeling-2D-LC-MS/MS quantitative proteomic strategy, PRM-based targeted proteomic method, quantitative dot blot analysis, and tissue immunohistochemistry method, was used to build a diagnostic signature and a metastatic signature (Fig. 1).

### Patients and HCs
A total of 359 CRC patients (242 males and 117 females; median age 59 years, min–max: 26–87 years) were recruited from the Cancer Hospital, Chinese Academy of Medical Sciences, from January 2015 to October 2018. All patients were pathologically diagnosed by two senior pathologists, and random morning midstream urine samples were collected prior to surgical operations or chemotherapy/radiotherapy. We excluded 26.7% of patients with postoperative disease; a pathological diagnosis of nontubular adenocarcinoma (mucinous adenocarcinoma, melanoma, signet-ring carcinoma, neuroendocrine carcinoma); other benign or malignant tumors; abnormal renal functions; receiving chemoradiotherapy; and a failure of QC of PRM (without signals in >40% of peptides) or dot blot analysis (CV > 20%). The 263 qualified patients were divided into CRC patients without metastases (NM, $n = 76$), with LNM ($n = 97$) and with distant metastasis (DM, $n = 90$) according to the pathology report (Fig. 1).

In addition, 298 urine samples from HCs (173 males and 125 females; median age 55 years, min–max: 23–78 years) were obtained from the Health Medical Center of the Cancer Hospital and PLA General Hospital from August 2014 to October 2018. The enrollment criteria for HC subjects were as follows: (1) the absence of benign or malignant tumors; (2) a qualified physical examination finding no dysfunction of vital organs and (3) normal renal function and without albuminuria. Nine HCs were excluded for QC of PRM (without signals in more than 40% of peptides) or dot blot analysis (CV > 20%) (Fig. 1). Table 1 lists the demographic and clinical characteristics of the 552 patients and HCs. After collection, urine samples were stored at -80 °C.

### Sample preparation
Before use, the thawed urine samples were centrifuged in a thermostatic centrifuge for 45 min at 5000 × g and 4 °C, and the supernatant was collected. For isobaric tandem mass tag (TMT) and PRM assays, urinary proteins were enriched via our developed Urimem method with some modifications[39]. Briefly, 40 ml of urine was diluted with 40 ml of 0.2 M $Na_2HPO_4$ buffer, and the mixture was passed through a nitrocellulose membrane (0.22 μm), which was placed onto a vacuum suction filter bottle (10 cm² filter area). After drying at 56 °C in an oven, the protein-bound membrane was cut into small pieces and placed in a 2 mL tube, to which 1.7 mL of acetone and 250 μL of 0.5% $NH_4HCO_3$ were added. After 10 min of intense vortexing, the tube was then incubated at 4 °C for 1 h, followed by centrifugation at 14,000 × g for 15 min. The precipitates were collected and air-dried (30 min) at room temperature. Afterward, 400 μL of lysis buffer (7 M urea, 2 M thiourea, 65 mM DTT, and 82 mM Tris) was added to resuspend the pellets, followed by intense vortexing for 10 min. The sample was centrifuged at 3500 × g for 30 min at 4 °C, and the supernatant was collected and then quantified by the Bradford assay.

The protein was digested by the filter-aided sample prep method. After digestion, peptides were extracted by a C18 extraction column and dried under a vacuum. The peptide concentration was further quantified by the BCA method.

### TMT labeling and 1D off-line separation
Nine randomly selected samples from the HC, NM, LNM, and DM groups were individually labeled with the 126, 127 N, 127 C, 128 N, 128 C, 129 N, 129 C, 130 N, and 130 C 10-plex TMT reagents according to the manufacturer's protocol (Thermo Fisher Scientific, Waltham, MA, USA). A mixed sample from all four groups was labeled with 131 TMT reagents. The labeled samples from each group were mixed individually. The pooled mixture of TMT-labeled samples was fractionated using a high-pH reversed-phase liquid chromatography (RPLC) column from Waters (4.6 mm × 250 mm, Xbridge C18, 3 μm). The samples were loaded onto the column in buffer A1 ($H_2O$, pH = 10). The elution gradient was 5–25% buffer B1 (90% acetonitrile, pH = 10; flow rate: 0.8 mL/min) for 48 min. The eluted peptides were collected at one fraction per minute. The 48 dried fractions were resuspended in 0.1% formic acid and pooled into 24 samples by combining fractions 1 and 25, 2 and 26, and so on.

### LC-MS/MS analysis
The fractionated labeled samples were analyzed using a self-packed RP C18 capillary LC column (75 μm × 100 mm, 1.9 μm). A total of 96 fractions from urinary peptide mixtures in the four groups were analyzed by LC-MS/MS. The gradient was eluted in 5–30% buffer B1 (0.1% formic acid, 99.9% $H_2O$; flow rate: 0.3 μL/min) for 45 min. Each sample was run 3 times. LTQ Orbitrap Fusion Lumos MS (Thermo Fisher Scientific) was used to acquire raw data through xCalibur3.1 software (Thermo Fisher Scientific). MS data were acquired using the following parameters: top speed data-dependent mode (3 s) per full scan, full scans acquired in Orbitrap at a resolution of 60,000, MS/MS scans with 32% normalized collision energy in HCD mode at a resolution of 15,000, charge state screening (excluding precursors with unknown charge state or +1 charge state) and dynamic exclusion (exclusion size list 500, exclusion duration 30 s).

### Database searching
The MS/MS spectra were searched against the SwissProt human database from the UniProt website (www.UniProt.org) using the Proteome

Discoverer software suite (v2.1, Thermo Fisher Scientific). Trypsin was chosen as the cleavage specificity with a maximum number of allowed missed cleavages of two. Carbamidomethylation of cysteine and TMT 10-plex labels were set as the fixed modifications, and the oxidation of methionine, deamidation of asparagine and glutamine, carbamyl of lysine and the peptide N-terminus were set as the dynamic modifications. The searches were performed using a peptide tolerance of 20 ppm and a product ion tolerance of 0.05 Da. As a filter, a 1% false-positive rate at the protein level was used, and each protein contained at least one unique peptide.

After filtering the results as described above, the peptide abundances in different reporter ion channels of the MS/MS scan were normalized. The protein abundance ratio was based on unique peptide results. Proteins with a technical CV over 0.3 or an interindividual CV over 0.6 within each group were excluded. Proteins with a fold change ≥ 1.5 between the NM, LNM, or DM group and the control group were considered differential proteins.

**PRM study design.** According to a previously published PRM study design[15,16], first, a pooled sample was obtained by mixing the same amount of digested peptide from each individual in all four groups. Second, the pooled sample was used to design the targeted LC-MS/MS method on 5600 Triple-TOF instruments through Analyst 2.0 software (AB Sciex). In addition, 1–3 unique peptides of each protein were selected for PRM analysis. The generated spectra were assigned to peptide sequences by spectral matching with the reference urine proteome spectrum library generated in our previous study[9]. The quantification of peptides with confirmed identity was performed based on the fragment ion intensity at the apex of the corresponding chromatogram. The peptides that could be identified in the PRM method design were included for further analysis. Third, the individual samples were analyzed. To estimate system stability during the whole analysis process, the pooled sample was used as a QC to observe the stability of the instrument signal. During the whole analysis process, QC was analyzed before and after all samples and among every 8–10 samples. To avoid system errors, samples were analyzed in random orders, and different groups of samples were interleaved analyzed. A total 82 samples and 20 QC samples were analyzed. Peptides with a QC CV of less than 0.6 and identified in more than 70% of the samples were retained for further quantitative analysis. The Pearson's correlation coefficient between each pair of QC samples was estimated to analyze the system stability during the analysis.

**PRM analysis.** For the urinary CRC and HC samples, each sample was analyzed with a C18 RP self-packed capillary LC column (75 μm × 100 mm). The eluted gradient was 5–30% buffer B2 (0.1% formic acid, and 99.9% ACN; flow rate: 0.3 μL/min) for 40 min. A Triple-TOF 5600 mass spectrometer was used to analyze eluted peptides from the LC. The MS data were acquired using the high-sensitivity mode with the following parameters: PRM mode, full scans acquired at a resolution of 40,000 and MS/MS scans at a resolution of 20,000, rolling collision energy, charge state screening (including precursors with +2 to +4 charge state), MS/MS scan range of 100–1800 m/z, and scan time of 100 ms. Each sample was run twice.

For the urinary samples of bladder cancer and renal cell cancer, each sample was analyzed on a C18 RP self-packed capillary LC column (75 μm × 500 mm). The elution gradient was 5–30% buffer B2 (0.1% formic acid, 99.9% ACN; flow rate of 0.5 μL/min) for 45 min. An LTQ Orbitrap Fusion Lumos instrument was used to acquire raw data. The data were acquired using the following parameters: PRM mode; full scans and MS/MS scans were acquired in Orbitrap at resolutions of 60,000 and 15,000, respectively; 32% normalized collision energy was acquired in HCD mode; and the isolation window was 4. Each sample was run once.

**PRM data analysis.** PRM data processing was performed with Skyline 20.1 software. All the results were imported into Skyline, the correct peaks were selected manually, and all the peptide results in all samples were exported. The total ionic chromatography (TIC) of the +2–+5 charges of each sample was extracted by Progenesis software. The abundance of each peptide of each sample was normalized with the TIC of the respective sample to correct the sample loading amount and MS signal intensity. The PRM results, including protein names and peak areas, were exported for further analysis, and the differential proteins between different groups were screened and compared using the TMT results.

**Generating predictors for CRC based on PRM data.** To generate urinary protein biomarker panels that distinguish CRC patients from HCs (diagnostic model) as well as CRC patients with metastasis (including regional lymph node metastatic and distant metastatic patients) from those without metastases (metastatic model), the protein expression abundance obtained from PRM data were normalized following the methodology described in a previous study[40].

First, we calculated the correlation matrix of protein expression abundance by Spearman's rank correlation to measure the intercorrelation between peptides. To screen the classifiers, the proteins with low expression similarity were used to calculate the importance of discriminating two classes in the diagnostic and metastatic models using the randomForest R package. To minimize randomness, 100 random forests consisting of 150,000 trees were computed to generate averaged mean decrease accuracy values for each protein. Mean decrease accuracy values were averaged for each protein among the 100 random forest replicates. Next, the

area under the ROC curve (AUC) was calculated as a performance measure. The overlapping proteins with the top 10 highest mean decrease accuracy values and AUC values were used for subsequent analyses. That is, the AUC of the combination of any two proteins was computed. The representative proteins that showed the highest AUC and the strongest complementarity with other proteins were chosen as the most relevant features. The above analyses were performed using the R statistical environment.

**Quantitative dot blot analysis of urinary proteins.** Dot blot analysis of urinary protein was performed using a Whatman Minifold I 96-well dot blot array system (GE Healthcare, Chicago, IL, USA) according to the manufacturer's instructions. Briefly, the PVDF membrane was first immersed in methanol for 20 min and then in PBS for 10 min. Then, the dot blot apparatus was assembled, and 500 μl of diluted urine samples or standards in PBS were loaded into each well. The recombinant human proteins for CORO1C (Cat. No. RY-02857), ARPC5 (Cat. No. H00010092-P01), RAD23B (Cat. No. H00005887-P01), GSPT2 (Cat. No. H00023708-P01) and NDN (Cat. No. H00004692-P01) were purchased from RunYu BioTech. Inc. (Shanghai, China) and Abnova (Taiwan, China). Next, the vacuum was applied to filter the sample through the PVDF membrane. Thereafter, the membrane was blocked with 10% skim milk in PBS and probed with primary antibodies against CORO1C (Cat. No. H00023603-M02, Abnova), ARPC5 (Cat. No. sc-166760, Santa Cruz Biotech, Dallas, TX, USA), RAD23B (Cat. No. A1034, ABclonal Technology, Woburn, MA, USA), GSPT2 (Cat. No. 12989-1-AP, Proteintech Group Inc., Rosemont, IL, USA) and NDN (Cat. No. sc-101224, Santa Cruz Biotech). Following intensive washing, the membranes were developed using an enhanced chemiluminescence detection reagent (Thermo Fisher Scientific, Waltham, MA, USA) and visualized with the ImageQuant LAS4000 system (GE Healthcare, Chicago, IL, USA) with the intensity adjusted to avoid saturation of the spots.

Spot intensities were measured and corrected to the background with ImageJ software. The raw concentration of each sample was calculated by standard curves and then corrected by several samples that were common to each study and run on each blot. Additionally, urinary creatinine concentration was quantified using the Creatinine Parameter Assay Kit (R&D Systems, Minneapolis, MN, USA). The relative absorbance units of each protein were normalized to that of urinary creatinine excretion.

**Immunoassay verification.** To facilitate the comparison with commonly used clinical marker serum CEA, in the 434 urine samples detected by immunoassay, we excluded 122 samples without pre-operative CEA results (101 HCs and 21 CRC) from the subsequent modeling analysis. The rest 312 samples, including 154 HCs and 158 CRC patients (41 NM, 62 LNM, and 55 DM), were used to verify the performance of urinary protein diagnostic and metastatic signatures.

For the diagnostic model, DM group was excluded because it is at uncurable stage. A total of 257 samples, consisting of 154 HCs, 41 CRC NM patients, and 62 CRC LNM patients, were divided into a training set (67% of data set) and validation set (33% of the dataset) using the block randomization method. Briefly, samples from control and disease groups were sorted by age and sex and were numbered sequentially into 20 blocks. Next, the samples were sorted and divided into training and validation sets based on the random number (0 or 1) that was generated for each sample in each block. Thus, 171 subjects were enrolled in the training set (103 HC vs. 68 CRC) and 86 in the validation set (51 HCs vs. 35 CRC patients). The model was constructed using logistic regression by MedCalc 15 (New York, NY, USA). In the training phase, early- and intermediate-stage CRC patients were referred to as the disease group, while HC subjects were referred to as the control group. The diagnostic model was constructed by binary logistic regression and further validated using 10-fold cross-validation and an external validation set. The method of binary logistic regression is "enter" and variables were entered if P value <0.05.

For the metastasis model, all patients were enrolled (41 NM vs. 62 LNM and 55 DM CRC). LNM and DM patients were assigned to the metastasis groups, and NM patients were assigned as nonmetastatic CRC. The metastasis model was trained using the leave-one-out cross-validation method based on averaged neural network (avNNet) algorithm using caret R package to evaluate the performance for metastatic risk stratification.

**Immunohistochemistry staining.** Sixteen tissue microarrays (TMAs) of colon cancer were purchased from Shanghai Outdo Biotech Co., Ltd. (Shanghai, China) and SuperBiotek Co., Ltd. (Shanghai, China). Among them, twelve and two TMAs contained overall survival and RFS follow-up information, respectively. All TMAs were used for the immunohistochemistry staining of CORO1C, whereas nine of sixteen TMAs were used for RAD23B and ARPC5 analyses due to availability.

After deparaffinization and rehydration, the TMAs were immersed in methanol containing 0.3% hydrogen peroxide for 10 min to block endogenous peroxidase. Heat-induced antigen retrieval was performed in a water bath for 30 min in 0.1 M sodium citrate buffer (pH 6.0). After washing, TMAs were incubated overnight with anti-CORO1C (Cat No. TA349821; OriGene Technologies, Inc, Rockville, MD, USA), anti-RAD23B (Cat No. A1034), or anti-ARPC5 (Cat No. sc-166760) antibodies at 4 °C. Staining was performed using the Prolink-2 Plus HRP rabbit

polymer detection kit (Golden Bridge International Inc., Bothell, WA, USA) according to the manufacturer's instructions. The images were captured using Aperio ScanScope CS software (Vista, CA).

The results were evaluated separately by two independent pathologists. The staining intensity and area were quantified as described previously[41]. A staining index between 0 and 12 was achieved by multiplying the extent of positivity and intensity. For CORO1C, a staining index was used in which 0–3 was considered negative, 4–7 was weakly positive and 8–12 was strongly positive. For RAD23B, a staining index was used in which 0–3 was considered negative, 4–8 was weakly positive and 9–12 was strongly positive. For ARPC5, a staining index was used in which 0–6 was considered negative, 7–9 was weakly positive and 10–12 was strongly positive.

**Statistical analyses**. Pattern recognition analysis (PCA and OPLS-DA) was performed using SIMCA 14.0 (Umetrics, Sweden) software. Unsupervised clustering was performed using the MetaboAnalyst tool (www.metaboanalyst.ca). Complete clustering with the Euclidean distance was performed on the group average protein quantitation data. Mann–Whitney rank test, one-way ANOVA and Kruskal–Wallis test with Dunn's multiple comparison test was used for statistical analyses of quantitative data with GraphPad Prism software (v7; San Diego, CA, USA). Chi-square analysis was used to compare qualitative data with IBM SPSS software (v18; Chicago, IL, USA). The ROC curves were plotted using SigmaPlot 14.0 (Systat Software Inc, San Jose, CA, USA). The Kaplan–Meier curve followed by log-rank analysis was performed to compare survival curves using survminer R package. The other analyses previously mentioned are described above. Statistical significance was defined as a two-sided $P$ value of <0.05.

**Reporting summary**. Further information on research design is available in the Nature Research Reporting Summary linked to this article.

## Data availability

The authors declare that the data supporting the findings of this study are available within Supplementary Information and Supplementary Data files 1–16. And Source Data are also provided with this paper. The MS proteomics data have been deposited to the ProteomeXchange Consortium (http://proteomecentral.proteomexchange.org) via the iProX partner repository[42] (http://iprox.cn) with the dataset identifier PXD032291 and IPX0002679000.

## Code availability

The in-house scripts used to generate some data in the paper are deposited in the GitHub (https://github.com/scshaochen/PRMColonCancer).

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

## Acknowledgements

We are especially grateful to Professor Y. Gao of Beijing Normal University for technical support regarding urine sample collection and helpful suggestions for this study. We thank Professors W. Cui and Z. Zhou of Cancer Hospital of CAMS for administrative support for sample collection, and members of the Zhao lab for their help. This work was supported by grants from the National Natural Science Foundation of China (81872033 to X.Z., 82073327 to Y.S., 82170524 to W.S. and 31901039 to X.L.), National Key R & D Program (2017YFC0906603, 2018YFC1313101 and 2016YFC0901403 to X.Z.), the State Key Project for Infectious Diseases (2017ZX10203205-003 to X.Z.), Non-profit Central Research Institute Fund of CAMS (2018RC310011 to W.S.), Beijing Medical Research (2018-7 to W.S.), and the CAMS Innovation Fund for Medical Sciences (2016-I2M-1-001 and 2019-I2M-1-003 to X.Z., 2021-I2M-1-066 to Y.S. and 2021-I2M-1-016 to W.S.).

## Author contributions

X.Z., W.S., Y.S., and Z.G. designed all studies and discussed the results. M.C., L.Y., J.L., Y.Z., J.G., X.X. L.T., L.W., and P.N. collected and prepared urine samples. Z.G., W.S., H.S. and X.L. performed urinary proteomics analysis, including TMT labeling, LC-MS/MS and PRM analyses. C.S., W.S., Y.S., Z.G., X.L., Z.J and L.Y. performed database searching and bioinformatics and statistical analyses. F.L. and S.Z. performed immunohistochemistry staining and pathological analysis. L.Y. performed a quantitative dot blot analysis of urinary proteins. Z.Z., Y-F.Z., M.W., and Y.H. selected patients and collected their clinical information. Y.S. wrote the manuscript with input from all authors. X.Z. and W.S. guided the work, analyzed data, and wrote the manuscript.

## Competing interests

The authors declare no competing interests.
