## [Peer Review File · Nature Communications]

Noninvasive urinary protein signatures associated with colorectal cancer diagnosis and metastasisEditorial Note: Parts of this Peer Review File have been redacted as indicated to maintain the confidentiality of unpublished data.

REVIEWER COMMENTS

Reviewer #1 (Remarks to the Author):

This study established novel urinary protein biomarker for detecting colorectal cancer (CRC) and predicting metastatic risk of CRC. Among 41 proteins that were selected as urinary biomarker candidates in the discovery stage (N=36) using TMT labeling MS approach, a signature comprising CORO1C, ARPC5 and RAD23B was a urinary diagnostic biomarker for CRC and a signature comprising CORO1C, RAD23B, GSPT2 and NDN were a urinary predictive biomarker for metastatic CRC in the verification stage (N=82) using targeted MS analysis. Finally, these signatures showed consistent diagnostic performance in both training and validation sets of the validation stages (N=434) using dot analysis. Combined these signatures with serum CEA, AUCs of diagnostic signature were 0.93 in the training set and 0.89 in the validation set, and those of metastatic signature were 0.85 in the training set and 0.74 in the validation set. Only CORO1C expression in the CRC tissues was also analyzed, which was associated with advanced stage. These findings were also confirmed in vitro and in vivo studies.

This study showed validity of established urinary biomarker using multidisciplinary approach. As the results, I think the data form this study would be convincing. Although there are a few concerns as shown below, this study could generally identify novel urinary biomarker for CRC.

Major:

1. In terms of diagnostic biomarker, early diagnosis is essential. In the view of early detection, discrimination between NM and HC is the most important, as shown in supplementary Fig. 5 C and K. These figures should be emphasized as main Figure and described in manuscript, with sensitivity, specificity and accuracy.

2. The critical limitation of this study includes only 22 stage I CRC patients among 263 CRC patients. And there are no stage 0 CRCs. Although it may be difficult to show significance, we just want to see detectability for stage I CRC. In addition, the author needs to describe small amount of early-stage CRC patients as limitation in Discussion.

Although the author described NM as early stage, it is wrong. Only stage 0/I is early stage. This description should be also amended because stage II is localized cancer, but advanced stage.

3. Only CORO1C was analyzed in the tissue validation. How about others? I am not sure why you focused on only CORO1C. In addition, CORO1C is associated with advanced stage in the tissue, in vitro and in vivo studies, suggesting that it is not suitable for early detection. At least, I want to see tissue expression of other diagnostic proteins including ARPC5 and RAD23B.

4. In terms of metastatic risk analysis each stage cohort, does it compare between NM and LNM/DM? (or between NM and DM?)

It is unclear. Please clarify more.

Minor:

Although the protein name is spelled as ARPC5 in Figures, it is described as APRC5 in manuscript. I think ARPC5 would be right? Please correct the spelling.

Reviewer #2 (Remarks to the Author):

In the current study, Yulin Sun et al reported urinary protein signatures of colorectal cancer. The identification of 8000 proteins in human urine is significant. The study offer novel urinary protein biomarkers and potential interventional targets to reliably detect CRC. However, there are some shortages throughout the paper.

1.It is mentioned that urine proteins have good diagnostic value and have been verified comprehensively. But whether these changes in protein expression are related to urinary diseases, I think it needs to be validated.

2.What is the role of CORO1C in diseases of the urinary system? Research on its downstream pathways is not sufficient.

3.Some grammar and typing errors needed to be corrected. There are some markings on the picture that are not clear.

Reviewer #3 (Remarks to the Author):

The manuscript entitled “Noninvasive urinary protein signatures associated with colorectal cancer diagnosis and metastasis” by Sun et al. describes the discovery, verification and validation of biomarkers for diagnosing colorectal cancer (CRC) and stratification of different stages of CRC. The initial discovery and verification is performed using LC/MS-based proteomics, followed by validation using various other biochemical and biological assays ranging from dot blots to xenograph models. First of all, I would like to congratulate the team around Y. Sun on the tremendous amount of work that they have carried out. It is a very thorough study which crossed a lot of Ts and dots most of the Is. Having said that, this manuscript is a classic example of less is more. The authors are providing so much detail and data, that the manuscript in its current form is barely comprehensible. I dare to say that half of the figure panels and supplementary figures can be removed without taking away from the bigger story. The manuscript in its current form reminds me of an attempt to make a photo album documenting the previous year without being willing to make (hard) choices as to which photo is good/important and should be shown and which can be removed from the list.

While the authors are exceeding detailed about the results from any (minor) experiment or analysis, they are surprisingly sparse when it comes to explain the choice of (analytical) methods, interpretation of data. Although there are various examples throughout the text, the most striking example is the transition from the discovery to the verification. While in the discovery section, hundreds of proteins (see additional comment below) are discussed as being dysregulated, the verification stage continues with 41 proteins. It is nowhere to be found HOW these 41 proteins were selected, although such selection is crucial in the biomarker discovery and validation process.

In summary, the manuscript in its current form cannot be accepted. It needs a major re-write before acceptance can be considered.

Additional comments/issues (semi-arbitrary order):

Line 44/45: please rephrase “Moreover, over 50% of CEA-negative metastatic patients were correctly predicted by the metastatic signature generated for metastatic risk.” The triple use of the word “metastatic” makes it hard to comprehend. Furthermore, the signature was generated to predict or stratify the risk.

In the introduction, the authors refer to various metastases, without clearly defining which makes it hard for the people not in the field. For instance, thy talk about local, distant, synchronous distant and about delayed liver metastases. Later, they also talk about lymph node metastases, without clarifying whether distant or local. In summary, cleaning up this paragraph and being more succinct with the different metastases classes would be helpful.

Line 73-75: the authors list 5 percentage values/ranges for 4 cancer stages. Please clarify.

Line 115/117: it is not clear which cohorts and subgroups the authors are referring to.

Line 122: Why are the number of proteins identified in the different groups so widely different? Please clarify/explain/discuss.

Line 124: It is not clear why the authors apply a 60% cut-off for the interindividual CV. Aren't the proteins with big CV potentially the relevant ones if they are e.g. very high in the disease group and very low in the control group?!

Line 140/141: It seems that the authors only applied a cut-off of 1.5fold difference, without any statistical considerations. Given that such approach is unusual by now, it should be explained and justified. Furthermore, please clarify whether mean or median fold difference was used.

Line 170: Without details about study design, number of standards and how they are interspersed between the real samples, this number is not very useful.

Line 166: Explanation and justification for the choice of the PRM approach might be useful.

Line 199+: it is not clear how the different CRC groups are combined in order to prepare a single ROC for the diagnostic model and one for the staging model. Please clarify this important aspect in the main text.

Figure 4A – E (top panels) are not necessarily helpful as they just show a random/representative subset. Just eyeballing, e.g. 4B and E does not show much of a difference. As such, the bottom panels are much more helpful.

Line 225: please clarify/explain the TNM and M staging.

Line 245-247: "Due to the small sample size of the NM group, 27 NM samples were duplicated twice, and a total of 81 NM and 78 metastatic CRC samples were used to train the model." Do the authors mean "duplicated TWICE" or just "duplicated". Also, the mere act of duplicating samples has to be well justified and explained as it makes the statistics difficult as the samples are not independent any longer. How was that accounted for?

Line 236+: please clarify how the samples were split into the two groups? What was done to prevent any potential selection bias?

Minor issue: the authors tend to write APRC5, although ARPC5 is the correct protein acronym.

Reviewer #4 (Remarks to the Author):

The authors identified five protein biomarkers from urine that can distinguish healthy controls from Colorectal cancer samples using various protein quantification methods and machine learning. They were able to demonstrate these new biomarkers are complementary to traditional serum-based biomarkers. To validate these new biomarkers' sensitivity and specificity, the authors tested two independent cohorts on two data platforms. The authors further used IPA analysis to understand the pathway enrichment for proteins with different control and cancer levels. Overall, the findings are important for the field and can impact clinical practice if validated on a large sample scale. However, there are several significant concerns on the details of machine learning and statistical analysis.

Major concerns:

It is unclear if the performance observed in this study is due to overfitting. The training and testing process is not well described. How were the training and testing samples selected? What if the selection process is randomized?

In line 179, the author said, "multilevel machine learning was used...". What does it mean? Pearson correlation is not a machine learning method.

Machine learning models trained on one data platform cannot be directly applied to another data platform. There are three data platforms involved in the study. How was the model learned from one platform transfer to another platform. Was cross-validation used to understand if there is any overfitting?

Minor concerns:

The authors can be more specific when describing the results. For example, in lines 199-200, it is not clear whether 86% sensitivity is computed on training or testing without checking the figure and previous section.

PCA or OPLS-DA. A clear separation of OPLS-DA on the discovery set is expected since DA is a supervised learning method. This leads to a false impression that HC, NM, LNM, and DM are easily separable. However, this is not the case in a larger cohort. I would recommend using the unsupervised PCA to replace the supervised LS-DA.

Reviewer #5 (Remarks to the Author):

In this study, Sun and coll. have investigated urine-derived protein biomarkers as a non-invasive approach for diagnosing colorectal (CRC) cancer and evaluating metastasis. They characterized diagnostic and metastatic signatures for predicting the risks of cancer and metastasis which both include CORO1C. CORO1C is an actin network component that was further tested on colorectal cell lines as a potential mediator of cell migration and invasion.

The work for developing the urinary protein signatures for CRC is interesting and was performed using state of the art discovery and validation stage proteomic approaches. There are previous studies showing the feasibility of using urinary proteins as biomarkers for detecting CRC cancer and patients developing metastasis but less detailed than the current one. Discovery stage was performed using urine samples obtained from controls, stages I-II (NM), stages III (LNM) and stage IV (DM) cancers using a relatively low number followed by a verification stage using targeted MS on 13 to 25 samples per group. Five biomarkers were then validated by dot blot to establish the final signature and expression of CORO1C was confirmed on tissues samples.

In parallel, the effect of CORO1C was evaluated on CRC cancer cell lines for its promoting effect on cell growth, migration, and invasion. Here also, there are previous studies showing that CORO1C regulates cell proliferation, survival, migration, and invasion but on various cell types other than CRC.

There are several issues that need however to be addressed in the manuscript.

1- While previous studies have mainly used urinary markers for evaluating and monitoring colorectal metastasis, the present study also included CRC diagnosis, which at first sight appears original. However, there are inconsistencies with this approach. First, it is difficult to rationalize that stage I and II CRC lesions can release significant amounts of specific proteins in the circulation to be detected in the urine. Second, in the diagnosis model, it is noteworthy that stage IV lesions are normally excluded for evaluating sensitivity and specificity of biomarkers since the screening approach is to identify CRC cancers that are at curable stages. Including stage IV artificially improves the statistics (lines 199-200). Third, and most important, CEA is not a biomarker used for diagnosis of CRC in the clinic. Comparisons of urinary-derived signature stats with CEA are therefore irrelevant. The claim "Thus our urinary diagnostic signature is a potent biomarker panel for the detection of early-stage CRC with better accuracy than serum CEA alone" (lines 398-399) should be omitted. At present it is the FIT that

is used for CRC diagnosis and that should be used for comparison.

2- Rectal cancer frequently generates metastasis in the thorax instead of the liver. Was this considered for the DM group?

3- line 284: The choice of 90% for specificity appears arbitrary. 95% would be more adequate in order to compare with other methods.

4- lines 294-305: Showing CORO1C tissue expression in metastasis would strengthen the association of this marker with the metastatic process.

5- lines 307-317: The lack of effect of CORO1C overexpression or silencing on CRC cell growth, apoptosis or colony formation is surprising considering what has been reported in the literature with other cell models as well as the data shown with xenografts where proliferation is enhanced. This discrepancy should be addressed.

6- lines 331-335/Figure 5O: It is not clear that all of the proteins listed are "markedly increased" along with CORO-1C overexpression. Same for phosphorylation. For instance, both p-FAK and FAK appear to be increased in the same range so there is no net increase in phosphorylation, Same for c-Jun while there is a net decrease in SRC and JNK phosphorylation. Please revise.

7- lines 319-329: The basic xenograft model is useful for studying tumor cell growth at ectopic sites but other models addressing more specifically tumor invasion and metastasis in vivo would help to better define the involvement of CORO1C in CRC progression. For instance, direct injection of the cells into the mouse circulation via distinct sites (arterial, venous, portal vein, etc.) could be informative of the potential of organ colonization.

Reviewer #1 (Remarks to the Author):

This study established novel urinary protein biomarker for detecting colorectal cancer (CRC) and predicting metastatic risk of CRC. Among 41 proteins that were selected as urinary biomarker candidates in the discovery stage (N=36) using TMT labeling MS approach, a signature comprising CORO1C, ARPC5 and RAD23B was a urinary diagnostic biomarker for CRC and a signature comprising CORO1C, RAD23B, GSPT2 and NDN were a urinary predictive biomarker for metastatic CRC in the verification stage (N=82) using targeted MS analysis. Finally, these signatures showed consistent diagnostic performance in both training and validation sets of the validation stages (N= 434) using dot analysis. Combined these signatures with serum CEA, AUCs of diagnostic signature were 0.93 in the training set and 0.89 in the validation set, and those of metastatic signature were 0.85 in the training set and 0.74 in the validation set. Only CORO1C expression in the CRC tissues was also analyzed, which was associated with advanced stage. These findings were also confirmed in vitro and in vivo studies.

This study showed validity of established urinary biomarker using multidisciplinary approach. As the results, I think the data form this study would be convincing. Although there are a few concerns as shown below, this study could generally identify novel urinary biomarker for CRC.

Authors' response: We sincerely appreciate all of the positive comments.

Major:

The reviewer's comment 1: *In terms of diagnostic biomarker, early diagnosis is essential. In the view of early detection, discrimination between NM and HC is the most important, as shown in supplementary Fig. 5 C and K. These figures should be emphasized as main Figure and described in manuscript, with sensitivity, specificity and accuracy.*

Authors' response: We thank the reviewer for the suggestion. We reorganized the figures and supplementary figures in the revised manuscript. The original supplementary Fig. 6c is shown as the updated Fig. 4b. Meanwhile, the sensitivity,

specificity, accuracy and AUC are described in Lines 244-245 in the Results section. It should be noted that according to the other reviewer's opinion, we removed the comparative analysis of diagnostic model and serum CEA. Hence, the original Supplementary Fig. 6K is not included in our revised manuscript.

The reviewer's comment 2: The critical limitation of this study includes only 22 stage I CRC patients among 263 CRC patients. And there are no stage 0 CRCs. Although it may be difficult to show significance, we just want to see detectability for stage I CRC. In addition, the author needs to describe small amount of early-stage CRC patients as limitation in Discussion.

Although the author described NM as early stage, it is wrong. Only stage 0/I is early stage. This description should be also amended because stage II is localized cancer, but advanced stage.

Authors' response: We thank the reviewer for the comment and suggestion. We agree with the reviewer's opinion that only stage 0/I disease is early stage disease. Therefore, we reanalyzed the predictive accuracy for stage I using the diagnostic signature consisting of CORO1C, ARPC5 and RAD23B. For PRM analysis, 11 stage I CRC patients and 25 healthy controls (HCs) were enrolled. The diagnostic signature achieved a sensitivity, specificity and AUC of 72.7%, 88.0% and 0.782 for stage I CRC, respectively (Revised Supplementary Figure 2B). For dot blot analysis, 11 stage I CRC patients and 24 randomly selected HCs were enrolled. The diagnostic signature yielded a sensitivity, specificity and AUC of 90.9%, 83.3% and 0.879, respectively (Revised Supplementary Figure 3A). The diagnostic power of the diagnostic model to discriminate stage I CRC from HCs was not significantly decreased compared with that between the NM group and HCs. We have added these results to the "Verification of differential urinary proteins using PRM-based targeted MS" (Line 202-204) and "Performance of urinary protein diagnostic and metastatic signatures" (Line 241-243) subsections of the Results section.

Revised Supplementary Fig. 2B (partial). ROC curves of the diagnostic signature to discriminate CRC patients with stage I disease from healthy controls in PRM analysis.

Revised Supplementary Fig. 3A. ROC curves of the diagnostic signature to discriminate CRC patients with stage I disease from healthy controls in dot blot analysis.

To further validate the effect of the diagnostic signature in early-stage CRC patients, we collected new urine samples from 8 CRC patients containing stage 0 (n=4) and stage I (n=4) disease, and 9 sex- and age-matched HCs. The levels of CORO1C, ARPC5 and RAD23B in urine were quantified using the PRM targeted proteomic analysis. The diagnostic signature yielded an AUC, sensitivity and specificity of 0.833, 75.0% and 100% (Fig. A).

Figure A. ROC curves of the diagnostic signature to discriminate new collected urine samples from patients with CRC stage 0/stage I disease from HCs in the PRM analysis.

Collectively, the above results suggested that our diagnostic signature was discriminative for the diagnosis of early-stage CRC. However, because only a small number of early-stage CRC patients were enrolled, the accuracy of the diagnostic model for detecting stage 0/I CRC needs to be further validated using larger cohorts, and even with cohorts from multiple centers. In addition, we have added the limitation in the discussion section: “Because only small amount of early-stage CRC patients were enrolled, the accuracy of diagnosis model for detecting stage I CRC needs to be further validated using larger cohorts, even from multiple centers (please to see in Lines 459-463).”

The reviewer’s comment 3: Only CORO1C was analyzed in the tissue validation. How about others? I am not sure why you focused on only CORO1C. In addition, CORO1C is associated with advanced stage in the tissue, in vitro and in vivo studies, suggesting that it is not suitable for early detection. At least, I want to see tissue expression of other diagnostic proteins including ARPC5 and RAD23B.

Authors’ response: We thank the reviewer for the comment and suggestion. Among the five screened urinary biomarkers, CORO1C was the protein shared by the diagnostic and metastatic signatures, and its aberrant expression and functions was not investigated in CRC yet. It was found that CORO1C regulates actin cytoskeleton

dynamics and may promote tumor invasion and metastasis. It is necessary for the release of inactive RAC1 from the nonprotrusive membrane and the activation and redistribution of RAC1 to a protrusive tip.^{1,2} Therefore, in the previous version of our manuscript, we focused on only CORO1C in the tissue validation.

As the reviewer mentioned, our tissue, *in vitro* and *in vivo* studies showed that CORO1C was associated with tumor progression and advanced stage. However, numerous studies have indicated that the acquisition of metastatic capacity is an early event in tumorigenesis, even in certain preneoplastic lesions^{3,4}. Therefore, metastasis-related proteins could be upregulated at an early stage of CRC, even in precancerous tissues. Our PRM and dot blot analyses confirmed that CORO1C was a discriminative urinary marker for CRC.

Currently, there have been no reports on CORO1C, RAD23B and ARPC5 expression in early-stage CRC. According reviewer's suggestion, we added tissue validation for RAD23B and ARPC5 with an immunohistochemical (IHC) staining to investigate the expression pattern of CORO1C, RAD23B and ARPC5 in precancerous diseases and early-stage CRC. We also performed an IHC staining assay in several tissue microarrays that contained normal colon or rectum epithelium (n=8), low-grade intraepithelial neoplasia (LGIN) lesions (n=21), high-grade intraepithelial neoplasia (HGIN) lesions (n=41) and CRC. The results indicated that the expression levels of CORO1C, RAD23B and ARPC5 were showed relative weak in the normal mucosal epithelium. Moreover, the expression levels of CORO1C and RAD23B were gradually increased with the progression of LGIN to HGIN. The level of ARPC5, however, was dramatically increased at the LGIN stage. In the adjacent noncancerous tissues, CORO1C, RAD23B and ARPC5 staining remained relatively weak and was significantly lower than HGIN tissues. Positive cytoplasmic immunostaining for CORO1C, RAD23B and ARPC5 was observed in 55.7% (535/961), 84.0% (420/500) and 85.8% (429/500) of the CRC tumor tissues, respectively, showing significant upregulation compared with that in the adjacent noncancerous and precancerous tissues (Fig. B).

Figure B. Elevated expression of three diagnostic biomarkers in tissues in precancerous lesions was revealed by an immunohistochemical staining assay. (A) Representative immunohistochemistry images of CORO1C, RAD23B and ARPC5 expression in normal colorectal mucosa, low-grade intraepithelial neoplasia (LGIN), high-grade intraepithelial neoplasia (HGIN), paracarcinoma normal tissues (PN) and CRC tumors (T). Original magnification: upper panel, $\times 40$; bottom panel, $\times 200$. (B-D) Distribution of CORO1C (B), RAD23B (C) and ARPC5 (D) protein levels, as determined by immunohistochemistry, in each group. The short dotted line represents the median value. *, **, ***, **** denote a *P* value of <0.05 , <0.01 , <0.001 and <0.0001 respectively.

Furthermore, we compared the expression of CORO1C, RAD23B and ARPC5 in different clinical TNM stages with paracarcinoma normal tissues. As shown in Fig. C, the expression of the three proteins was significantly elevated in the patients with stage I disease (all $P < 0.0001$). The level of CORO1C was not associated with the TNM stage ($P = 0.1078$), whereas CRC patients with stage II-IV disease had higher expression levels of ARPC5 ($P = 0.0021$) and RAD23B ($P = 0.0010$) than those with stage I disease.

Figure C. The expression levels of CORO1C, RAD23B and ARPC5 in paracarcinoma normal and tumor tissues with different TNM stages.

Taken together, our results indicated that CORO1C, RAD23B and ARPC5 were suitable for early detection because their expression levels were significantly increased at the precancerous stage and associated with malignant progression in CRC. These results were shown and reorganized in Fig. 5 and Supplementary Fig. 4 in our revised manuscript.

References

- Williamson, R. C., Cowell, C. A., Reville, T., Roper, J. A., Rendall, T. C. & Bass, M. D. Coronin-1C Protein and Caveolin Protein Provide Constitutive and Inducible Mechanisms of Rac1 Protein Trafficking. *J Biol Chem* **290**, 15437-15449 (2015).
- Williamson, R. C., et al. Coronin-1C and RCC2 guide mesenchymal migration by trafficking Rac1 and controlling GEF exposure. *J Cell Sci* **127**, 4292-4307 (2014).
- Hu, Y., Yu, X., Xu, G., & Liu, S. Metastasis: an early event in cancer progression. *Journal of cancer research and clinical oncology* **143**, 745–757 (2017).
- Lambert, A. W., Pattabiraman, D. R., & Weinberg, R. A. Emerging Biological Principles of Metastasis. *Cell*, **168**, 670–691 (2017).

The reviewer's comment 4: *In terms of metastatic risk analysis each stage cohort, does it compare between NM and LNM/DM? (or between NM and DM?)*

It is unclear. Please clarify more.

Authors' response: We thank the reviewer for the comment. In terms of metastatic risk analysis among each stage cohort, we compared the NM group with the LNM or DM groups. In the PRM verification assay, the performance of the metastatic signature for discriminating the three CRC groups was analyzed in pairwise

comparisons. The signature differentiated the NM group from the LNM and DM groups with AUCs of 0.723 and 0.827, respectively (revised Supplementary Table 3d, revised Supplementary Fig. 2c). These detailed analyses are included in Lines 210–212.

In the dot blot verification assay, for the discrimination of LNM or DM from NM, the metastatic signature yielded AUCs of 0.710 and 0.826, respectively (revised Supplementary Table 4c, revised Supplementary Fig. 3b). These analyses are included in Lines 253-255.

Minor:

The reviewer's comment 5: *Although the protein name is spelled as ARPC5 in Figures, it is described as APRC5 in manuscript. I think ARPC5 would be right? Please correct the spelling.*

Authors' response: We thank the reviewer for highlighting this issue. ARPC5 is the correct spelling. We have fixed these typographical errors in the revised manuscript.

Reviewer #2 (Remarks to the Author):

In the current study, Yulin Sun et al reported urinary protein signatures of colorectal cancer. The identification of 8000 proteins in human urine is significant. The study offer novel urinary protein biomarkers and potential interventional targets to reliably detect CRC. However, there are some shortages throughout the paper.

Authors' response: We sincerely appreciate the reviewer's comments.

The reviewer's comment 1: *It is mentioned that urine proteins have good diagnostic value and have been verified comprehensively. But whether these changes in protein expression are related to urinary diseases, I think it needs to be validated.*

Authors' response: We thank the reviewer for the suggestion. The question you mentioned is very important for evaluating the specificity of these five candidate biomarkers for CRC. To evaluate whether these changes in protein expression are

related to urinary diseases, we quantified additionally the urinary protein concentrations of CORO1C, RAD23B, ARPC5, GSPT2 and NDN in patients with bladder cancer (n=20) and renal cell carcinoma (n=22) as well as in sex- and age-matched healthy controls (n=20 and 20, respectively) using a PRM targeted proteomic strategy. The Kruskal-Wallis test was used to evaluate the significance of these proteins among different groups. The results showed that there was no significant difference between renal cell carcinoma or bladder cancer patients and healthy controls in the concentrations of these five proteins in urine (revised Supplementary Fig. 5 and revised Supplementary Table 3e, 3e). Furthermore, to evaluate whether the expression changes of these five CRC biomarkers are related to benign urinary diseases, we searched the Ingenuity Pathway Analysis (IPA) biomarker database, and we did not find a relationship between these five proteins and benign urinary diseases. Whether the five proteins are related to other urinary tumors or diseases needs to be further explored in the future.

Revised Supplementary Fig. 5. Urinary levels of CORO1C, RAD23B, ARPC5, GSPT2, and NDN in urological tumors. The urine samples of patients with bladder cancer (a) and renal cell carcinoma (b) as well as sex- and age-matched healthy controls were measured using a PRM targeted proteomic strategy. The median and quartile values in each group of individuals are shown as thick red dotted lines and thin purple dotted lines, respectively. PRM, parallel reaction monitoring; HC, healthy controls; BC, bladder cancer; RCC, renal cell carcinoma; ns, not significant.

It was concluded that the protein expression of these three urinary proteins was not related to urinary disease. We have added this information to the Discussion section in Line 439-448.

The reviewer's comment 2: What is the role of CORO1C in diseases of the urinary system? Research on its downstream pathways is not sufficient.

Authors' response: We thank the reviewer for the comments. One part of the comment has already been replied to in a question above. Briefly, our new PRM targeted proteomic analysis showed that the protein concentrations of CORO1C in urine were not significantly different between patients with bladder cancer or renal cell carcinoma and healthy controls (revised Supplementary Fig. 5). Subsequently, we investigated the role of CORO1C in diseases of the urinary system through a literature review. Only a very recent study reported that CORO1C contributed to the progression of bladder cancer based on TCGA RNA sequencing data.⁵ The overexpression of CORO1C was positively associated with poor prognosis, epithelial mesenchymal transition (EMT) and tumor-infiltrating neutrophils. However, this study did not perform mechanistic studies or measure protein expression in urine samples of patients with bladder cancer. Further studies are needed to assess the role of CORO1C in other diseases of the urinary system.

Regarding the second comment, few studies have explored the roles of CORO1C in wound healing, protrusion formation, cell proliferation, cytokinesis, endocytosis, axonal growth, secretion, migration and invasion.⁶⁻⁹ In the present study, we found that CORO1C enhances the invasion and metastasis of CRC cells via a new pathway, integrin/FAK/SRC relative signaling pathways. Additionally, we also found that RAD23B interacted with CORO1C to promote the colocalization of RAC1 and CORO1C to the lateral edges of CRC cells to form invasive protrusions and invadopodia, which enhanced the migratory and invasive abilities of CRC cells.¹⁰

According to the editor's comments, we removed our limited functional assays on CORO1C to focus on the diagnostic/prognostic value of the urine protein signatures. Thus, the functional studies of on CORO1C are not shown in our revised manuscript.

References

5. Wang C, et al. Novel Potential Biomarkers Associated With Epithelial to Mesenchymal Transition and Bladder Cancer Prognosis Identified by Integrated Bioinformatic Analysis. *Front Oncol* **10**, 931 (2020).
6. Rosentreter A., et al. Coronin 3 involvement in F-actin-dependent processes at the cell cortex. *Exp Cell Res* **313**, 878-895 (2007).
7. Xavier C. P., et al. Phosphorylation of CRN2 by CK2 regulates F-actin and Arp2/3 interaction and inhibits cell migration. *Sci Rep* **2**, 241 (2012).
8. Castagnino A., et al. Coronin 1C promotes triple-negative breast cancer invasiveness through regulation of MT1-MMP traffic and invadopodia function. *Oncogene* **37**, 6425-6441 (2018).
9. Fan L., Wei Y., Ding X. & Li B. Coronin3 Promotes Nasopharyngeal Carcinoma Migration And Invasion By Induction Of Epithelial-To-Mesenchymal Transition. *Onco Targets Ther* **12**, 9585-9598 (2019).
10. Li J, et al. Cytoplasmic RAD23B interacts with CORO1C to synergistically promote colorectal cancer progression and metastasis. *Cancer Lett* **516**, 13-27 (2021).

The reviewer's comment 3: Some grammar and typing errors needed to be corrected. There are some markings on the picture that are not clear.

Authors' response: We thank the reviewer for these comments. We carefully reorganized and revised the figures, supplementary figures and figure legends in our revised manuscript to make them clearer. Furthermore, the revised manuscript has been proofread by a native-English speaking editor at American Journal Experts (AJE) (<http://www.aje.com/>), an editing company, for checking grammar, vocabulary, and style. The relevant editing certificate number is DCAD-9A53-1354-DE68-552B.

Reviewer #3 (Remarks to the Author):

The reviewer's comment 1: The manuscript entitled "Noninvasive urinary protein signatures associated with colorectal cancer diagnosis and metastasis" by Sun et al. describes the discovery, verification and validation of biomarkers for diagnosing colorectal cancer (CRC) and stratification of different stages of CRC. The initial discovery and verification is performed using LC/MS-based proteomics, followed by validation using various other biochemical and biological assays ranging from dot

blots to xenograph models. First of all, I would like to congratulate the team around Y. Sun on the tremendous amount of work that they have carried out. It is a very thorough study which crossed a lot of Ts and dots most of the Is. Having said that, this manuscript is a classic example of less is more. The authors are providing so much detail and data, that the manuscript in its current form is barely comprehensible. I dare to say that half of the figure panels and supplementary figures can be removed without taking away from the bigger story. The manuscript in its current form reminds me of an attempt to make a photo album documenting the previous year without being willing to make (hard) choices as to which photo is good/important and should be shown and which can be removed from the list.

Authors' response: We thank the reviewer for the helpful comment and suggestion. We reorganized the figures, supplementary figures and supplementary tables in our revised manuscript to remove unnecessary supplementary figures or figure panels and combine Supplementary Tables (e.g., original Supplementary Fig. S3, Fig. S4, Fig. S5, Fig. S1a, Fig. S2a, Fig. S2c, Fig. S6a, and Fig. S6f-S6m). Original Supplementary Table 1a-1f was combined in the new Supplementary Table 1a. Moreover, according to the editor's suggestion, we removed the in vitro and in vivo assays on CORO1C to focus on the diagnostic/prognostic value of the urine protein signatures in the revised manuscript. Thus, original Fig. 5 and Supplementary Fig. S7 were also removed from our revised manuscript. Additionally, we added some new experiments based on the reviewers' comments. Thank you again for your constructive advice. We hope that our revised manuscript addresses your concerns.

The reviewer's comment 2: *While the authors are exceeding detailed about the results from any (minor) experiment or analysis, they are surprisingly sparse when it comes to explain the choice of (analytical) methods, interpretation of data. Although there are various examples throughout the text, the most striking example is the transition from the discovery to the verification. While in the discovery section, hundreds of proteins (see additional comment below) are discussed as being dysregulated, the verification stage continues with 41 proteins. It is nowhere to be*

found HOW these 41 proteins were selected, although such selection is crucial in the biomarker discovery and validation process.

Authors' response: We thank the reviewer for these comments. In this study, a widely used biomarker discovery workflow, a “triangular” strategy,^{11,12} was adopted to discover and develop new urinary protein markers for CRC (Fig. 1). In the discovery stage, quantitative proteomic technology was applied to identify hundreds of differentially expressed proteins in a small sample size (n = 36). In the verification stage, the candidate biomarkers (100s) were selected for verification in a larger sample size (n = 82) using targeted proteomics methods (MRM/PRM). Only some of the differential proteins that showed similar trends in the discovery stage were selected for the next validation stage.¹³⁻¹⁵ In the validation stage, five biomarkers were validated in an independent larger scale sample size (n = 434). Our entire study took approximately 6–7 years to complete.

The parallel reaction monitoring (PRM) approach is a widely used targeted proteomics technology for proteomic analysis.¹⁶⁻¹⁸ It is based on high-resolution and accurate mass spectrometry that permits the simultaneous detection of all target product ions in one, concerted high resolution mass analysis. PRM yielded a wider dynamic range, better quantification precision, and achieved better linearity than traditional selected reaction monitoring.^{19,20}

Specifically, in our study, a total of 718 differential proteins (including 581 CRC-related and 226 CRC metastasis related differential proteins) were identified in the discovery stage. In addition, 112 and 54 proteins showing gradient increasing or decreasing tendencies along with CRC progression, respectively, were selected for PRM verification. For each protein, 1~3 unique peptides were selected for PRM analysis. Among them, 77 proteins (173 peptides) were identified by PRM method design and were analyzed at the PRM stage. Consequently, 66 proteins (107 peptides) could be quantified using the PRM method. In addition, 41 proteins (66 peptides) were verified with trends consistent with those of the TMT approach in the four groups.

We added these descriptions in Lines 162-175 in the Results section and Lines 129-147 in the Supplementary Materials and Methods section.

The choice and the detailed workflow of the PRM analytical method and the detailed interpretation of the data analysis are described in detail in the comments below.

References

11. Rifai, N., Gillette, M. A., & Carr, S. A. Protein biomarker discovery and validation: the long and uncertain path to clinical utility. *Nature biotechnology*, **24**, 971–983 (2006).
12. Geyer, P. E., Holdt, L. M., Teupser, D., & Mann, M. Revisiting biomarker discovery by plasma proteomics. *Molecular systems biology*, **13**, 942 (2017).
13. Zhou, Q., et al. Quantitative proteomics identifies brain acid soluble protein 1 (BASP1) as a prognostic biomarker candidate in pancreatic cancer tissue. *EBioMedicine*, **43**, 282–294 (2019).
14. Guo, J., et al. A Comprehensive Investigation toward the Indicative Proteins of Bladder Cancer in Urine: From Surveying Cell Secretomes to Verifying Urine Proteins. *Journal of proteome research*, **15**, 2164–2177 (2016).
15. Chu, H. W., Chang, K. P., Hsu, C. W., Chang, I. Y., Liu, H. P., Chen, Y. T., & Wu, C. C. Identification of Salivary Biomarkers for Oral Cancer Detection with Untargeted and Targeted Quantitative Proteomics Approaches. *Molecular & cellular proteomics*, **18**, 1796–1806 (2019).
16. Hutton, J. E., et al. Oncogenic KRAS and BRAF Drive Metabolic Reprogramming in Colorectal Cancer. *Molecular & cellular proteomics*, **15**, 2924–2938 (2016).
17. Martinez-Garcia, E., et al. Targeted Proteomics Identifies Proteomic Signatures in Liquid Biopsies of the Endometrium to Diagnose Endometrial Cancer and Assist in the Prediction of the Optimal Surgical Treatment. *Clinical cancer research*, **23**, 6458–6467 (2017).
18. Johnson, E., et al. Large-scale proteomic analysis of Alzheimer's disease brain and cerebrospinal fluid reveals early changes in energy metabolism associated with microglia and astrocyte activation. *Nature medicine*, **26**, 769–780 (2020).
19. Peterson, A. C., Russell, J. D., Bailey, D. J., Westphall, M. S., & Coon, J. J. Parallel reaction monitoring for high resolution and high mass accuracy quantitative, targeted proteomics. *Molecular & cellular proteomics*, **11**, 1475–1488 (2012).
20. Aebersold, R., & Mann, M. Mass-spectrometric exploration of proteome structure and function. *Nature*, **537**, 347–355 (2016).

The reviewer's comment 3: *In summary, the manuscript in its current form cannot be accepted. It needs a major re-write before acceptance can be considered.*

Additional comments/issues (semi-arbitrary order):

Line 44/45: please rephrase "Moreover, over 50% of CEA-negative metastatic patients were correctly predicted by the metastatic signature generated for metastatic

risk.” The triple use of the word “metastatic” makes it hard to comprehend.

Furthermore, the signature was generated to predict or stratify the risk.

Authors’ response: We thank the reviewer for the comment and suggestion. The signature refers to the urinary protein panel for metastasis risk stratification. Thus, we replaced the sentence with “Moreover, the generated metastatic signature for risk stratification correctly predicted over 50% of CEA-negative metastatic patients.” in Lines 44-45..

***The reviewer’s comment 4:** In the introduction, the authors refer to various metastases, without clearly defining which makes it hard for the people not in the field. For instance, thy talk about local, distant, synchronous distant and about delayed liver metastases. Later, they also talk about lymph node metastases, without clarifying whether distant or local. In summary, cleaning up this paragraph and being more succinct with the different metastases classes would be helpful.*

Authors’ response: We thank the reviewer for the comment and suggestion. The “localized” stage refers to invasive CRC that has penetrated the wall of the colon or rectum with no sign that it has spread outside of the colon or rectum. The regional stage was defined as invasive CRC invaded through the wall of the colon or rectum, and had spread to nearby structures and/or lymph nodes. Distant spread or metastasis referred to invasive CRC that had spread to distant parts of the body such as the liver, lungs, peritoneum (lining of the abdomen), or ovaries. The above staging system is more often used by cancer registries.

The most widely used cancer staging system in the clinic is the TNM staging system. A detailed description of the significance of the TNM system is provided below (please see the reply to the comment 14 on page 27). However, in general, the local stage corresponds to stages I and II in the TNM system. The regional stage corresponds to stage III, while distant metastasis corresponds to stage IV. Lymph node metastasis usually refers to regional lymph node metastasis; thus, it is in the regional stage or stage III in the TNM system.

Furthermore, distant metastasis can occur at any time during the progression of

CRC. In the clinic, synchronous metastasis was defined as distant metastases detected before or within 6 months of the initial diagnosis of primary CRC²¹.

"Metachronous" metastasis was defined as the occurrence of distant metastasis within six months after initial diagnosis.

To refine the description in this paragraph and make it more succinct and clear, we streamlined the two staging systems used in the original version into the TNM staging system in Lines 56-65.

Reference

21. Pantaleo, M. A., Astolfi, A., Nannini, M., Paterini, P., Piazzzi, G., Ercolani, G., Brandi, G., Martinelli, G., Pession, A., Pinna, A. D., & Biasco, G. (2008). Gene expression profiling of liver metastases from colorectal cancer as potential basis for treatment choice. *British journal of cancer*, 99(10), 1729–1734.

The reviewer's comment 5: Line 73-75: the authors list 5 percentage values/ranges for 4 cancer stages. Please clarify.

Authors' response: We thank the reviewer for the comment. In Lines 73–75 (original manuscript), we listed 5 percentage values or ranges, which referred in turn to colorectal cancer stages I, II, III, and IV diseases and recurrence. To clarify the description, we revised the sentence “the sensitivity of serum CEA for patients with stage I, II, III, IV diseases and recurrence is 4–11%, 25–30%, 38–44%, 65% and 50–71%, respectively” in Lines 73-75.

The reviewer's comment 6: Line 115/117: it is not clear which cohorts and subgroups the authors are referring to.

Authors' response: We thank the reviewer for the comment. In Lines 115–117, the three CRC patient cohorts refer to the NM (patients without metastases), LNM (patients with lymph node metastasis), and DM (patients with distant liver metastasis) cohorts. The three subgroups within each cohort refer to the samples used for TMT, PRM and dot blot analyses. To clarify this point, we revised the sentence using the annotations in parentheses in Lines 115-117.

The reviewer's comment 7: Line 122: Why are the number of proteins identified in the different groups so widely different? Please clarify/explain/discuss.

Authors' response: We thank the reviewer for the comment. It is indeed like the reviewer mentioned, there is a general phenomenon that protein amounts identified in tumor tissues are significantly greater than those in adjacent normal tissues in proteomics studies.²²⁻²⁵ This effect might be related to the presence of a large number of heterotemporal and spatially expressed proteins in the tumor. In addition, aberrant signaling pathways and metabolism in tumors may also contribute.

Specifically, in previous large-scale urinary proteomic studies, the number of quantified proteins was also widely different in the healthy control and different disease groups. The results were similar to those in our study. For example, Zhang et al. reported that the average number of proteins quantified in the urine of the different tumor groups and the healthy control group ranged from 1014 to 1842 (Table A).²⁶ Tian et al. reported that the average number of proteins quantified in the urine of the control group, non-COVID-19 disease group, and COVID-19 group were 4131, 3869, and 2837, respectively (Table B).²⁷ These phenomena might be associated with the characteristics of the disease status and individual variation,²⁸ and further exploration is needed.

Table A. Average number of proteins quantified in different disease groups²²

	Sample size	Number of proteins quantified
Healthy controls	33	1552±432
Lung cancer	33	1014±218
Bladder cancer	17	1146±166
Cervical cancer	25	1198±149
Colorectal cancer	22	1201±268
Esophageal cancer	14	1153±178
Gastric cancer	47	1311±254
Benign lung diseases	40	1842±585

Table B. Average number of proteins quantified in different disease groups²²

	Sample size	Number of proteins quantified
--	-------------	-------------------------------

Health control	10	4131±288
Non-COVID-19 disease control	13	3869±600
COVID-19	14	2837±528

References

22. Jiang, Y., et al. Proteomics identifies new therapeutic targets of early-stage hepatocellular carcinoma. *Nature* **567**, 257-261 (2019).
23. Gillette, M. A., et al. Proteogenomic Characterization Reveals Therapeutic Vulnerabilities in Lung Adenocarcinoma. *Cell* **182**, 200-225 e235 (2020).
24. Xu, J. Y., et al. Integrative Proteomic Characterization of Human Lung Adenocarcinoma. *Cell* **182**, 245-261 e217 (2020).
25. Zhou, Y., et al. Proteomic signatures of 16 major types of human cancer reveal universal and cancer-type-specific proteins for the identification of potential therapeutic targets. *Journal of hematology & oncology*, **13**, 170 (2020).
26. Zhang, C., et al. Urine Proteome Profiling Predicts Lung Cancer from Control Cases and Other Tumors. *EBioMedicine*, **30**, 120–128 (2018).
27. Tian, W., et al. Immune suppression in the early stage of COVID-19 disease. *Nature communications*, **11**, 5859 (2020).
28. Virreira Winter, S., et al. Urinary proteome profiling for stratifying patients with familial Parkinson's disease. *EMBO molecular medicine*, **13**, e13257 (2021).

The reviewer's comment 8: Line 124: *It is not clear why the authors apply a 60% cut-off for the interindividual CV. Aren't the proteins with big CV potentially the relevant ones if they are e.g. very high in the disease group and very low in the control group?!*

Authors' response: We thank the reviewer for the comment. It is generally accepted that proteins with narrow interindividual CVs could serve as potential urinary biomarkers.²⁹⁻³¹ We first calculated the distribution of the interindividual CV of the four groups. The results showed that the disease groups displayed slightly a higher interindividual CV than the HC groups (Fig. D and Table C). The data from Zhang et al.'s study also showed that the urine interindividual CV was slightly higher in different cancer groups than in the control group (Table D).³² It should to be noted that because the data in Zhang et al.'s work were generated with a label-free method, the overall variations were significantly higher than those in our study using the TMT method.

Table C. The interindividual CVs in the four groups

	HC	CRC-NM	CRC-LNM	CRC-DM
--	----	--------	---------	--------

Median interindividual CV	0.193	0.260	0.276	0.215
% of individual CV<0.6	96.39%	94.09%	92.10%	97.81%

Figure D. The interindividual CV distribution in the four groups

Table D. The median interindividual CV in the control group and different tumor groups in the data generated by a label-free method²⁹

	Healthy controls	Lung cancer	Bladder cancer	Cervical cancer	Colorectal cancer	Esophageal cancer	Gastric cancer	Benign lung diseases
Median CV	1.20	1.50	1.20	1.33	1.44	1.31	1.32	1.05

Therefore, to identify candidate biomarkers of CRC more accurately, we excluded proteins with high interindividual variability. We choose a 60% cutoff for the interindividual CV (exclusion of proteins with approximately the top 5% of interindividual CV in each group). We added the above explanation in Line 125 in the revised manuscript: “exclusion of proteins with approximately the top 5% interindividual CV”.

References

29. Nedelkov, D., Kiernan, U. A., Niederkofler, E. E., Tubbs, K. A., & Nelson, R. W. Population proteomics: the concept, attributes, and potential for cancer biomarker research. *Molecular & cellular proteomics*, **5**, 1811–1818 (2006).
30. Sun, W., et al. Dynamic urinary proteomic analysis reveals stable proteins to be potential biomarkers. *Proteomics. Clinical applications*, **3**, 370–382 (2009).
31. He, W., et al. A stable panel comprising 18 urinary proteins in the human healthy population. *Proteomics*, **12**, 1059–1072 (2012).

32. Zhang, C., et al. Urine Proteome Profiling Predicts Lung Cancer from Control Cases and Other Tumors. *EBioMedicine*, **30**, 120–128 (2018).

The reviewer's comment 9: Line 140/141: It seems that the authors only applied a cut-off of 1.5fold difference, without any statistical considerations. Given that such approach is unusual by now, it should be explained and justified. Furthermore, please clarify whether mean or median fold difference was used.

Authors' response: We thank the reviewer for the comment. In this study, a widely used biomarker discovery workflow, a “triangular” strategy,^{33,34} was adopted to discover and develop new urinary protein markers for CRC (Fig. 1). In the discovery stage, discovery proteomic technology was used to identify hundreds or thousands of differentially expressed proteins in a small sample size (~10). The purpose of this stage was to screen as many disease-related differentially expressed proteins as possible; thus, relatively relaxed criteria were used.^{33,34} Due to the small sample size, statistical analysis was not possible. Therefore, only the fold change cutoff without statistical analysis was used at this stage.^{35,36} In the verification stage, 10s of candidate biomarkers were verified in a larger sample size (10s-100s) using a targeted proteomic method. In the validation stage, several biomarkers were validated in an even larger sample size (100s-1000s) using immunoassays.^{33,34} In the verification and validation stages, statistical analysis was used to validate the reliability of the candidate biomarkers.^{35,36}

Therefore, in the present study, at the discovery stage, we used 2D-LC-MS/MS and tandem mass tag (TMT)-labeled tandem mass spectrometry technology to analyze the urinary proteome of CRC in a small sample size (9 participants in each group). We only applied a cutoff of a 1.5-fold difference from the mean to screen CRC-related differentially expressed proteins. In the verification and validation stages, we used PRM and dot plot methods to validate the candidate biomarkers in larger independent sample sizes (82 and 434 samples), respectively. More stringent statistical criteria ($P < 0.05$ in the Mann-Whitney rank test or Kruskal-Wallis test followed by a Dunn's multiple comparisons test) were used.

We clarified the description in our revised manuscript in Lines 136-138: “The pairwise differential urinary proteins between NM, LNM or DM and HC were defined using a criterion of a mean fold change \geq 1.5”.

References

33. Rifai, N., Gillette, M. A., & Carr, S. A. Protein biomarker discovery and validation: the long and uncertain path to clinical utility. *Nature biotechnology*, **24**, 971–983 (2006).
34. Geyer, P. E., Holdt, L. M., Teupser, D., & Mann, M. Revisiting biomarker discovery by plasma proteomics. *Molecular systems biology*, **13**, 942 (2017).
35. Chu, H. W., Chang, K. P., Hsu, C. W., Chang, I. Y., Liu, H. P., Chen, Y. T., & Wu, C. C. Identification of Salivary Biomarkers for Oral Cancer Detection with Untargeted and Targeted Quantitative Proteomics Approaches. *Molecular & cellular proteomics*, **18**, 1796–1806 (2019).
36. Yeh, C. C., Hsu, C. H., Shao, Y. Y., Ho, W. C., Tsai, M. H., Feng, W. C., & Chow, L. P. Integrated Stable Isotope Labeling by Amino Acids in Cell Culture (SILAC) and Isobaric Tags for Relative and Absolute Quantitation (iTRAQ) Quantitative Proteomic Analysis Identifies Galectin-1 as a Potential Biomarker for Predicting Sorafenib Resistance in Liver Cancer. *Molecular & cellular proteomics*, **14**, 1527–1545 (2015).

The reviewer's comment 10: Line 170: Without details about study design, number of standards and how they are interspersed between the real samples, this number is not very useful.

Authors' response: We thank the reviewer for the comment. The selected differential proteins were verified in 82 samples by PRM. According to previously published PRM study design,³⁷⁻⁴¹ first, a pooled sample was analyzed in data-dependent acquisition (DDA) tandem MS mode to generate reference spectral library. Then, the pooled sample was used to design a targeted LC-MS/MS method. The generated spectra from PRM analysis were matched with the reference library. The quantification of peptides with a confirmed identity was performed based on the fragment ion intensity. The pooled sample was repeatedly analyzed in PRM method to evaluate the stability of the system. After that, the individual samples were analyzed in PRM method.

Based on the characteristics of the urinary proteomics, the following strategies were adopted in our study. First, a pooled sample was obtained by mixing the same amount of digested peptide from each individual in four groups. Second, we used the

pooled sample to design the targeted LC–MS/MS method. The targeted LC–MS/MS analyses in PRM mode were performed on 5600 Triple-TOF instruments. The generated spectra were assigned to peptide sequences by spectral matching with the reference urine proteome spectrum library generated in our previous study.⁴² The quantification of peptides with a confirmed identity was performed based on the fragment ion intensity at the apex of the corresponding chromatogram. Third, the individual samples were analyzed. To estimate the system stability during whole analysis process, the pooled sample was used as a quality control (QC) to observe the stability of the instrument signal (Fig. E). The QC samples were firstly analyzed in triplicate to ensure the system stability. The average Pearson correlation coefficient of the QC samples was 0.99, indicating that the system was stable. During the whole analysis process, QC was performed before and after all samples and in every 8-10 samples. To avoid system errors, samples were analyzed in random orders, and different groups of samples were interleaved analyzed. A total 82 samples and 20 QC samples were analyzed. The Pearson correlation coefficient between each pair of QC sample was estimated. The average Pearson correlation coefficient of the QC samples was 0.99, indicating that the system was stable during the analysis. The above description was added in Lines 129-147 in the Supplementary Materials and Methods section.

Figure E. The workflow of our PRM analysis

References

37. Peterson, A. C., Russell, J. D., Bailey, D. J., Westphall, M. S., & Coon, J. J. Parallel reaction monitoring for high resolution and high mass accuracy quantitative, targeted proteomics. *Molecular & Cellular Proteomics*, **11**, 1475–1488 (2012).
38. Khristenko, N. A., Larina, I. M., & Domon, B. Longitudinal Urinary Protein Variability in Participants of the Space Flight Simulation Program. *Journal of Proteome Research*, **15**, 114–124 (2016).
39. Hutton, J. E., et al. Oncogenic KRAS and BRAF Drive Metabolic Reprogramming in Colorectal Cancer. *Molecular & Cellular Proteomics*, **15**, 2924–2938 (2016).
40. Martinez-Garcia, E., et al. Targeted Proteomics Identifies Proteomic Signatures in Liquid Biopsies of the Endometrium to Diagnose Endometrial Cancer and Assist in the Prediction of the Optimal Surgical Treatment. *Clinical Cancer Research*, **23**, 6458–6467 (2017).
41. Johnson, E., et al. Large-scale proteomic analysis of Alzheimer's disease brain and cerebrospinal fluid reveals early changes in energy metabolism associated with microglia and astrocyte activation. *Nature medicine*, **26**, 769–780 (2020).
42. Zhao, M., et al. A comprehensive analysis and annotation of human normal urinary proteome. *Scientific reports*, **7**, 3024 (2017).

The reviewer's comment 11: Line 166: Explanation and justification for the choice of the PRM approach might be useful.

Authors' response: We thank the reviewer for the comment. The parallel reaction

monitoring (PRM) approach is a widely used targeted proteomics technology for proteomic analysis. It is based on high-resolution and accurate mass spectrometry that permits the parallel detection of the entire MS2 spectrum in one, concerted high-resolution mass analysis. PRM yields a wider dynamic range, better quantification precision, and achieved better linearity than selected reaction monitoring (SRM).^{43,44} PRM technology has been widely used in targeted proteomic studies.⁴⁵⁻⁴⁷

Because PRM technology combines high quantification precision and a high throughput design, we chose the PRM approach for the verification stage to quantify dozens of proteins in 82 samples. We added justification for the choice of the PRM approach in the Results section in Lines 162-163..

References

43. Peterson, A. C., Russell, J. D., Bailey, D. J., Westphall, M. S., & Coon, J. J. Parallel reaction monitoring for high resolution and high mass accuracy quantitative, targeted proteomics. *Molecular & cellular proteomics*, **11**, 1475–1488 (2012).
44. Aebersold, R., & Mann, M. Mass-spectrometric exploration of proteome structure and function. *Nature*, **537**, 347–355 (2016).
45. Hutton, J. E., et al. Oncogenic KRAS and BRAF Drive Metabolic Reprogramming in Colorectal Cancer. *Molecular & cellular proteomics*, **15**, 2924–2938 (2016).
46. Martinez-Garcia, E., et al. Targeted Proteomics Identifies Proteomic Signatures in Liquid Biopsies of the Endometrium to Diagnose Endometrial Cancer and Assist in the Prediction of the Optimal Surgical Treatment. *Clinical cancer research*, **23**, 6458–6467 (2017).
47. Johnson, E., et al. Large-scale proteomic analysis of Alzheimer's disease brain and cerebrospinal fluid reveals early changes in energy metabolism associated with microglia and astrocyte activation. *Nature medicine*, **26**, 769–780 (2020).

The reviewer's comment 12: Line 199+: it is not clear how the different CRC groups are combined in order to prepare a single ROC for the diagnostic model and one for the staging model. Please clarify this important aspect in the main text.

Authors' response: We thank the reviewer for the comment. For diagnosis model construction, NM and LNM patients were combined in the CRC disease group (defined as dependent variable 1), and HC subjects were referred to as the control group (defined as dependent variable 0). Binary logistic regression was used to

construct a diagnosis model. The ROC curve indicates the discrimination ability of the model for the CRC group (dependent variable: 1) from the controls (dependent variable: 0).

For metastasis model construction, LNM and DM patients were combined in the metastasis group referred to as metastatic CRC (defined as dependent variable 1), and NM patients were referred to nonmetastatic CRC (defined as dependent variable 0). Binary logistic regression was used to construct a metastasis model. The ROC curve indicates the discrimination ability of the model for metastatic CRC (dependent variable: 1) from nonmetastatic CRC (dependent variable: 0). We have added this information to the “Statistical analyses” subsection of the Materials and Methods section in Lines 532-541.

The reviewer’s comment 13: Figure 4A – E (top panels) are not necessarily helpful as they just show a random/representative subset. Just eyeballing, e.g. 4B and E does not show much of a difference. As such, the bottom panels are much more helpful.

Authors’ response: We thank the reviewer for the suggestion. We deleted the top panel of Figure 4a-4e. The revised Figure 4 is included in our revised manuscript.

The reviewer’s comment 14: Line 225: please clarify/explain the TNM and M staging.

Authors’ response: We thank the reviewer for the comment. The TNM classification system for malignant tumors is a globally recognized standard for cancer staging that was developed and is maintained by the Union for International Cancer Control (UICC) and the American Joint Committee on Cancer (AJCC). The T stage describes the size of the primary tumor and the depths of invasion into the surrounding tissues. The N stage refers to the number of regional lymph nodes that are involved. The M stage describes the distant metastasis or spread of cancer from the primary tumor to other parts of the body, such as the liver and lungs.

The latest (8th) edition of the TNM system for colorectal cancer:

T (Tumor)

Tis: Refers to carcinoma in situ (also called cancer in situ).

T1: The tumor has grown into the submucosa, which is the layer of tissue underneath the mucosa or lining of the colon.

T2: The tumor has grown into the muscularis propria, a deeper, thick layer of muscle that contracts to force the contents of the intestines to move.

T3: The tumor has grown through the muscularis propria and into the subserosa, or it has grown into tissues surrounding the colon or rectum.

T4a: The tumor has grown into the surface of the visceral peritoneum, which means that it has grown through all of the layers of the colon.

T4b: The tumor has grown into or has attached to other organs or structures.

Node (N)

N0: There is no spread to regional lymph nodes.

N1a: Tumor cells are found in 1 regional lymph node.

N1b: Tumor cells are found in 2 or 3 regional lymph nodes.

N1c: There are nodules of tumor cells found in the structures near the colon that do not appear to be lymph nodes.

N2a: Tumor cells are found in 4 to 6 regional lymph nodes.

N2b: Tumor cells are found in 7 or more regional lymph nodes.

Metastasis (M)

M0: The disease has not spread to a distant part of the body.

M1a: The cancer has spread to 1 other part of the body beyond the colon or rectum.

M1b: The cancer has spread to more than 1 part of the body other than the colon or rectum.

M1c: The cancer has spread to the peritoneal surface.

The entire TNM stage of colorectal cancer combining the T, N, and M classifications is shown in Table E⁴⁸

Table E. Colorectal cancer clinical staging based on the 8th edition of the TNM system

Stage	T	N	M
--------------	----------	----------	----------

0	Tis	N0	M0
I	T1-2	N0	M0
IIA	T3	N0	M0
IIB	T4a	N0	M0
IIC	T4b	N0	M0
IIIA	T1-2	N1/N1c	M0
	T1	N2a	M0
IIIB	T3-4a	N1/N1c	M0
	T2-3	N2a	M0
	T1-2	N2b	M0
IIIC	T4a	N2a	M0
	T3-4a	N2b	M0
	T4b	N1-2	M0
IVA	Any T	Any N	M1a
IVB	Any T	Any N	M1b
IVC	Any T	Any N	M1c

Reference

48. Weiser M. R. AJCC 8th Edition: Colorectal Cancer. *Annals of surgical oncology* **25**, 1454–1455 (2018).

The reviewer’s comment 15: Line 245-247: “Due to the small sample size of the NM group, 27 NM samples were duplicated twice, and a total of 81 NM and 78 metastatic CRC samples were used to train the model.” Do the authors mean “duplicated TWICE” or just “duplicated”. Also, the mere act of duplicating samples has to be well justified and explained as it makes the statistics difficult as the samples are not independent any longer. How was that accounted for?

Authors’ response: We thank the reviewer for the comment. In the present study, there were 27 NM samples and 78 metastatic CRC samples in the training set for the metastasis risk stratification model. This unbalanced sample size between the two groups in the training set is referred to as the problem of class imbalance. In this case, standard classifiers tend to be overwhelmed by the majority class and ignore the minority class.⁴⁹ The relative lack of information on the minority concept leads to a poor recognition of these elements by the classification model.⁵⁰ To solve the problems of a small sample size and class imbalance to construct a more robust model, oversampling is the simplest statistical method for imbalanced data through

the replication of minority class examples.^{49,50}

Based on the oversampling method, 27 samples in the NM group in the training set were triplicated to generate the 81 samples in the present study. Thus, this method addressed the underrepresentation of the NM group.

Of course, oversampling has the shortcoming that it will increase the likelihood of overfitting. To assess the impact of sample replication on the trained model, we evaluated it with an independent validation set in which no sample replication was conducted (i.e., all samples were independent). The validation results confirmed that the metastatic model constructed with this method was stable and robust. We have added a brief description of oversampling in the “Statistical analyses” subsection of the Materials and Methods section in Lines 541-545.

References

- 49 Guo X., Yin Y., Dong C., Yang G. & Zhou G. On the Class Imbalance Problem. *In: Fourth International Conference on Natural Computation*. **4**, 192-201 (2008), doi: 10.1109/ICNC.2008.871.
- 50 Japkowicz N. Class imbalances: are we focusing on the right issue. *In: Workshop on Learning from Imbalanced Data Sets II*, **1723**, (2003). doi:10.1109/TKDE.2008.239

The reviewer’s comment 16: *Line 236+: please clarify how the samples were split into the two groups? What was done to prevent any potential selection bias?*

The authors’ response: We thank the reviewer for the comment. We used the block randomization method to split samples into training and validation sets. It is a commonly used method for randomization to prevent potential selection bias.^{51,52} We set blocks for randomization and balanced the number of subjects in each block. First, samples from each condition (control and cancer) were sorted by age and sex. Then the samples were numbered sequentially into 20 blocks. A random number between 0 and 1 was generated for each sample using the RAND function in Excel. Finally, the samples in each block were sorted by a random number and sequentially divided into training and validation sets. We have added this information to the “Statistical analyses” subsection of the Materials and Methods section in Lines 535-536.

References

51. Lim, C. Y., & In, J. Randomization in clinical studies. *Korean journal of anesthesiology* **72**, 221–232 (2019).
52. Suresh K. An overview of randomization techniques: An unbiased assessment of outcome in clinical research. *Journal of human reproductive sciences* **4**, 8–11(2011).

The reviewer's comment 17: Minor issue: the authors tend to write APRC5, although ARPC5 is the correct protein acronym.

Authors' response: We thank the reviewer for highlighting this issue. We have fixed these typographical errors in our revised manuscript.

Reviewer #4 (Remarks to the Author):

The authors identified five protein biomarkers from urine that can distinguish healthy controls from Colorectal cancer samples using various protein quantification methods and machine learning. They were able to demonstrate these new biomarkers are complementary to traditional serum-based biomarkers. To validate these new biomarkers' sensitivity and specificity, the authors tested two independent cohorts on two data platforms. The authors further used IPA analysis to understand the pathway enrichment for proteins with different control and cancer levels. Overall, the findings are important for the field and can impact clinical practice if validated on a large sample scale. However, there are several significant concerns on the details of machine learning and statistical analysis.

Authors' response: We sincerely appreciate all of the positive comments.

Major concerns:

The reviewer's comment 1: It is unclear if the performance observed in this study is due to overfitting. The training and testing process is not well described. How were the training and testing samples selected? What if the selection process is randomized?

Authors' response: We thank the reviewer for the comments. The training and testing samples were selected randomly to prevent any potential selection bias using

the block randomization method.^{53,54} We set blocks for randomization and balance the number of subjects in each block. First, samples from each condition (control and cancer) were sorted by age and sex. Then the samples were numbered sequentially in 20 blocks. A random number between 0 and 1 was generated for each sample using the RAND function in Excel. Finally, the samples in each block were sorted by a random number, and sequentially divided into training and validation sets. The models were constructed using the training set and further validated using both internal and external validation to prevent overfitting. The detailed results are described in the following questions. We have added the randomization method to the “Statistical analyses” subsection of the Materials and Methods section in Lines 535-536.

References

53. Lim, C. Y., & In, J. Randomization in clinical studies. *Korean journal of anesthesiology* **72**, 221–232 (2019).
54. Suresh K. An overview of randomization techniques: An unbiased assessment of outcome in clinical research. *Journal of human reproductive sciences* **4**, 8–11(2011).

The reviewer’s comment 2: In line 179, the author said, " multilevel machine learning was used....". What does it mean? Pearson correlation is not a machine learning method.

Authors’ response: We thank the reviewer for the comment. We agree with the reviewer’s opinion. We used Spearman’s rank correlation, the random forest algorithm and combined classifiers based on ROC analysis to define urinary protein signatures. To make the description more accurate, we replaced the text with “Multilevel analysis was used ...” in our revised manuscript in Line 178.

The reviewer’s comment 3: Machine learning models trained on one data platform cannot be directly applied to another data platform. There are three data platforms involved in the study. How was the model learned from one platform transfer to another platform. Was cross-validation used to understand if there is any overfitting?

Authors’ response: We thank the reviewer for the comment. There were three data

platforms involved in the study, including TMT labeling quantitative proteomic analysis, PRM-based targeted proteomic analysis and immunological dot blot analysis. First, TMT labeling quantitative proteomic analysis was an untargeted method used to discover differential proteins between CRC groups and healthy controls. Thus, no model was constructed with this platform. Second, PRM-based targeted proteomic analysis was used to further validate these differential proteins from the former TMT platform. We used a machine learning algorithm and statistical analysis to select potential biomarkers for CRC diagnosis and metastasis. Then we used ROC curves to primarily evaluate the performance of our selected urinary protein signatures for CRC diagnosis and metastasis risk stratification. Finally, dot blot analysis was performed based on independent large-scale urine samples to further validate the potential biomarkers. New CRC diagnostic and metastasis risk stratification models were constructed at this stage. Thus, although potential biomarkers were transferred among the three platforms, the model learned from the PRM platform and did not transfer to the dot plot platform.

In addition, internal cross-validation and external validation were performed to evaluate PRM and dot blot model robustness. As suggested by the reviewer, we have added the results to the revised manuscript:

For PRM data, a urinary protein signature consisting of CORO1C, APRC5 and RAD23B exhibited good accuracy for CRC diagnosis with an AUC of 0.858. Due to the small sample size (HC: 25; CRC-LNM and CRC-NM: 33), 1,000 bootstrap resamples were used to evaluate the extent of model “overfitting”.⁵⁵ The bias-corrected AUC was 0.802, indicating that there was no overfitting of the model. A classifier for CRC metastasis consisting of CORO1C, RAD23B, GSPT2 and NDN achieved an AUC of 0.784. The bias-corrected AUC using 1000 bootstrap resamplings was 0.737, indicating the high robustness of the model (Table F).

For dot blot data, internal 10-fold cross-validation and external independent validation were used to evaluate the model robustness. For CRC diagnosis, a urinary protein signature consisting of CORO1C, APRC5 and RAD23B achieved an AUC of 0.787. Internal 10-fold cross-validation was performed, and the validated AUC was

0.777. Furthermore, an independent sample set was used to validate the diagnostic accuracy of the model. The external validation achieved an AUC of 0.846. For metastasis prediction, a classifier consisting of CORO1C, RAD23B, GSPT2 and NDN achieved an AUC of 0.773. Internal 10-fold cross-validation was performed, and the validated AUC was 0.772. The external validation achieved an AUC of 0.679 (Table F).

We have added the results to the “Verification of differential urinary proteins using PRM-based targeted MS” subsections in Lines 196-197 and 207-208 and “Performance of urinary protein diagnostic and metastatic signatures” subsections of the Results section in Lines 234-238 and 250-253.

Table F. Summary of robustness of CRC diagnosis and metastasis model in PRM and dot blot validation

		AUC	Sensitivity	Specificity
PRM-diagnosis panel	Training set	0.858	0.758	0.88
	1000 bootstrap correction	0.802	0.712	0.82
PRM-metastasis panel	Training set	0.784	0.811	0.7
	1000 bootstrap correction	0.737	0.663	0.8
Dot blot-diagnosis panel	Training set	0.787	0.691	0.796
	10-fold cross validation	0.777	0.676	0.806
	External validation	0.846	0.743	0.843
Dot blot-metastasis panel	Training set	0.773	0.577	0.926
	10-fold cross validation	0.772	0.667	0.864
	External validation	0.679	0.487	0.929

Reference

55. Dwivedi, A. K., Mallawaarachchi, I., & Alvarado, L. A. Analysis of small sample size studies using nonparametric bootstrap test with pooled resampling method. *Statistics in medicine* **36**, 2187–2205 (2017).

Minor concerns:

The reviewer’s comment 4: *The authors can be more specific when describing the results. For example, in lines 199-200, it is not clear whether 86% sensitivity is computed on training or testing without checking the figure and previous section.*

Authors’ response: We thank the reviewer for the comment and suggestion. We

have clarified the description in the text. In the PRM stage, 88.0% specificity and 75.8% sensitivity were computed for the training set (According to the other reviewers' opinion, DM group need to be excluded in the diagnostic model. Thus, the values shown here were changed.). There was no external testing set for PRM, but 1,000 bootstrap resamplings were used to validate the model robustness. We have revised the text in the "Verification of differential urinary proteins using PRM-based targeted MS" subsection of the Results section in Lines 194-195.

The reviewer's comment 5: PCA or OPLS-DA. A clear separation of OPLS-DA on the discovery set is expected since DA is a supervised learning method. This leads to a false impression that HC, NM, LNM, and DM are easily separable. However, this is not the case in a larger cohort. I would recommend using the unsupervised PCA to replace the supervised LS-DA.

Authors' response: We thank the reviewer for the comment and suggestion. We agree with the reviewer's opinion and replaced the OPLS-DA results to PCA analysis in the revised Fig. 2a.

Revised Fig. 2a. Score plot of unsupervised principal component analysis (PCA) overview of urinary proteomics among the HC, NM, LNM, and DM groups.

Reviewer #5 (Remarks to the Author):

In this study, Sun and coll. have investigated urine-derived protein biomarkers as a non-invasive approach for diagnosing colorectal (CRC) cancer and evaluating metastasis. They characterized diagnostic and metastatic signatures for predicting the risks of cancer and metastasis which both include CORO1C. CORO1C is an actin network component that was further tested on colorectal cell lines as a potential mediator of cell migration and invasion.

The work for developing the urinary protein signatures for CRC is interesting and was performed using state of the art discovery and validation stage proteomic approaches. There are previous studies showing the feasibility of using urinary proteins as biomarkers for detecting CRC cancer and patients developing metastasis but less detailed than the current one. Discovery stage was performed using urine samples obtained from controls, stages I-II (NM), stages III (LNM) and stage IV (DM) cancers using a relatively low number followed by a verification stage using targeted MS on 13 to 25 samples per group. Five biomarkers were then validated by dot blot to establish the final signature and expression of CORO1C was confirmed on tissues samples. In parallel, the effect of CORO1C was evaluated on CRC cancer cell lines for its promoting effect on cell growth, migration, and invasion. Here also, there are previous studies showing that CORO1C regulates cell proliferation, survival, migration, and invasion but on various cell types other than CRC.

Authors' response: We sincerely appreciate the reviewer's comments.

There are several issues that need however to be addressed in the manuscript.

The reviewer's comment 1: *While previous studies have mainly used urinary markers for evaluating and monitoring colorectal metastasis, the present study also included CRC diagnosis, which at first sight appears original. However, there are inconsistencies with this approach. First, it is difficult to rationalize that stage I and II CRC lesions can release significant amounts of specific proteins in the circulation to be detected in the urine. Second, in the diagnosis model, it is noteworthy that stage IV lesions are normally excluded for evaluating sensitivity and specificity of biomarkers*

since the screening approach is to identify CRC cancers that are at curable stages. Including stage IV artificially improves the statistics (lines 199-200). Third, and most important, CEA is not a biomarker used for diagnosis of CRC in the clinic. Comparisons of urinary-derived signature stats with CEA are therefore irrelevant. The claim "Thus our urinary diagnostic signature is a potent biomarker panel for the detection of early-stage CRC with better accuracy than serum CEA alone" (lines 398-399) should be omitted. At present it is the FIT that is used for CRC diagnosis and that should be used for comparison.

Authors' response: We thank the reviewer for the comment and suggestion. We will reply to the three comments individually.

For the first comment, urine, as the filtrate of the blood, does not need to remain stable in composition; therefore, it is easier to accumulate molecules that reflect changes throughout the whole body, even early and small pathological changes.⁵⁶ Urinary proteomes have already been found to reflect early changes in various tumor-bearing animal models,^{57,58} even for predicting tumor formation and distinguishing early tumor-forming rats from non-tumor-forming rats.⁵⁹ In human specimens, it was reported that urinary protein markers showed high diagnostic accuracy for identifying hepatocellular carcinoma among high-risk hepatitis C (HCV) infection patients with a lead time of 2 years.⁶⁰ Two recent studies also showed that urinary protein markers were significantly elevated in patients with stage I non-small-cell lung cancer (NSCLC) and gastric cancer compared with healthy controls.^{61,62} Thus, urinary proteins can reflect early-stage alterations during tumorigenesis in several tumors.

To further confirm this in CRC, we first analyzed the discriminative power of the diagnostic panel between the NM group (CRC stage I and stage II) and the HC group based on PRM and dot blot data. As shown in the revised Fig. 3e and Fig. 4b, the diagnostic panel showed good performance for stage I and stage II CRC with AUCs of 0.800 and 0.800 in PRM and dot blot data, respectively. Moreover, at the tissue level, we also found that the expression of the three proteins was significantly elevated in patients with stage I and stage II disease (Fig. F; all $P < 0.0001$). This result suggested

that the alterations of these markers in urine levels may reflect their change at the tissue level.

Revised Fig. 3e (partial). ROC curves for the diagnostic model (NM+LNM vs. HCs) to discriminate the HC group from the NM group (stage I+ stage II) in PRM analysis.

Revised Fig. 4b (partial). ROC curves for the diagnostic model (NM+LNM vs. HCs) to discriminate the HC group from the NM group (stage I+ stage II) in dot blot analysis.

Figure F. Elevated expression of CORO1C, RAD23B and ARPC5 in tissue occurred in stage I and stage II CRC.

To further confirm that the urinary changes in these markers in early-stage CRC patients resulted from early pathological changes in tissues, we newly collected samples from 8 stage 0/I CRC patients (4 stage I/4 stage 0) and 9 sex- and age-matched healthy controls. The levels of CORO1C, ARPC5 and RAD23B in urine were quantified using the PRM targeted proteomic method. The diagnostic signature yielded an AUC, sensitivity and specificity of 0.833, 75.0% and 100%, respectively (Fig. G). Meanwhile, we performed immunohistochemistry assays in all eight early-stage CRC patients. They showed positivity for the expression of CORO1C, RAD23B and ARPC5 in the tumor tissues. Thus, these results further confirmed that early alterations in CRC can be reflected in urine. However, further exploration is needed on how early-stage CRC lesions release these proteins into the urine..

Figure G. ROC curves of the diagnostic signature to discriminate newly collected urine samples from patients with CRC stage 0/stage I from healthy controls in the PRM analysis.

For the second comment, we agree with the reviewer's opinion. In our revised manuscript, stage IV disease was excluded when evaluating the sensitivity and specificity of the diagnostic model. For the PRM data, the diagnostic signature consisting of CORO1C, APRC5 and RAD23B achieved 88.0% specificity and 75.8% sensitivity with an AUC of 0.858 (Fig. 3e, Supplementary Table 3c). For the dot blot data, the diagnostic model in the training set (CRC vs. HCs, n=171) had an AUC, sensitivity, and specificity of 0.787, 69.1%, and 79.6%, respectively (Fig. 4b, Supplementary Table 4b). In the validation set (51 HCs vs. 35 CRC, total n=86), the signature achieved an AUC, sensitivity, and specificity of 0.846, 74.3%, 86.3%, respectively (Supplementary Fig. 3a, Supplementary Table 4b). Thus, although stage IV patients were excluded, there was no significant decrease in the diagnostic performance of our diagnostic signature.

We have revised Fig. 3e, Fig. 4b, Supplementary Fig. S3a, Supplementary Table 3c and Supplementary Table 4b. The corresponding results were revised in the "Verification of differential urinary proteins using PRM-based targeted MS" subsection in Lines 194-195 and "Performance of urinary protein diagnostic and metastatic signatures" subsection in Lines 231-238 of the Results section

Revised Fig. 3e. ROC curves for the diagnostic model (NM+LNM vs. HCs) to discriminate the HC group from the CRC group (NM+LNM) for PRM analysis

Revised Fig. 4b. ROC curve for the diagnostic model in the training set to discriminate the HC group from the CRC group (NM+LNM) for dot blot analysis.

Revised Supplementary Fig. 3a. ROC curve for the diagnostic model in the validation set to discriminate the HC group from the CRC group (NM+LNM) for dot blot analysis.

For the third suggestion, we searched for the FIT results of the enrolled CRC patients and HCs according to the reviewer's suggestion. Overall, 51 HCs and 33 CRC patients in the training set and 22 HCs and 16 CRC patients in the validation set had recorded qualitative FIT results. The specificity and sensitivity of FIT was 100% and 66.7% in the training set and 100% and 50.0% in the validation set, respectively. FIT is a good indicator for CRC diagnosis, with high specificity in our samples. We further compared the diagnostic performance of FIT with the urinary signatures using data from patients with FIT results (diagnosis training set: HCs: 51; CRC: 33; diagnosis validation set: HCs: 22; CRC: 16). The AUC, sensitivity and specificity of the

panel were 0.812, 72.7% and 86.3% in the training set and 0.864, 68.7% and 95.5% in the validation set, respectively. The diagnostic sensitivity of the urinary signature was higher than that of FIT. The combination of FIT and urinary proteins in the diagnostic signature had a better diagnostic capability. In the CRC patients in the training and validation cohorts, FIT was positive in 22 (66.7%) and 8 (50.0%) patients, and the urinary signature increased the diagnostic power in an additional 7 (21.2%) and 7 (43.7%) patients. For FIT-negative patients, 63.6% (7 out of 11 CRC patients) of patients in the training set and 87.5% (7 out of 8 CRC patients) of patients in the validation set were correctly diagnosed by the diagnostic signature (revised Fig. 4d). In conclusion, FIT is a good indicator for CRC diagnosis with a high specificity. The urinary signature significantly improved the sensitivity and complemented the FIT. We appreciate the reviewer’s important comments. According to the suggestion, we replaced CEA with the FIT test for CRC diagnosis performance comparisons. We added these results to the “Urinary protein signatures complemented FIT and CEA for CRC diagnosis and metastasis” subsection of the Results section in Lines 258-270.

Revised Fig. 4d. Diagnostic power of the diagnostic signature in individuals who were misdiagnosed by the FIT test in the training and validation groups

References

56. Wu, J. & Gao, Y. Physiological conditions can be reflected in human urine proteome and metabolome. *Expert Rev. Proteom.* **12**, 623–636 (2015).
 57. Zhang, H. et al. Identification of urine protein biomarkers with the potential for early detection of lung cancer. *Sci. Rep.* **5**, 11805 (2015).

58. Ni, Y., Zhang, F., An, M., Yin, W. & Gao, Y. Early candidate biomarkers found from urine of glioblastoma multiforme rat before changes in MRI. *Sci. China Life Sci.* **61**, 982–987 (2018).
59. Wei, J., Ni, N., Meng, W., Huan, Y. & Gao, Y. Early urinary protein changes during tumor formation in a NuTu-19 tail vein injection rat model. *Sci Rep* **10**, 11709 (2020).
60. Abdalla, M. A. & Haj-Ahmad, Y. Promising Urinary Protein Biomarkers for the Early Detection of Hepatocellular Carcinoma among High-Risk Hepatitis C Virus Egyptian Patients. *J Cancer* **3**, 390-403 (2012).
61. Ma, Y. C., et al. Urinary malate dehydrogenase 2 is a new biomarker for early detection of non-small-cell lung cancer. *Cancer Sci* **112**, 2349-2360 (2021).
62. Shimura, T., et al. Novel urinary protein biomarker panel for early diagnosis of gastric cancer. *Br J Cancer* **123**, 1656-1664 (2020).

The reviewer's comment 2: *Rectal cancer frequently generates metastasis in the thorax instead of the liver. Was this considered for the DM group?*

Authors' response: We thank the reviewer for the comment. We agree with the reviewer's opinion. We counted the metastatic sites in the patients from the DM group (n=58). As shown in Table G, the liver was the most common metastatic site for both colon and rectal cancer, although there was a tendency for colon cancer to have a higher proportion of liver metastasis than rectal cancer (93.1% vs. 82.8%; $P=0.4227$). In contrast, rectal cancer had a slightly higher proportion of lung alone or lung and distal lymph node metastasis (10.3% vs. 0; $P=0.2368$) than colon cancer..

Table G. Anatomical site of metastases in the DM group (n=58)

	Colon cancer (n=29)		Rectal cancer (n=29)		P value
	Number	(%)	Number	(%)	
One site					
Only liver	21	72.4	17	58.6%	0.2692
Only lung	0	0	1	3.4%	-
Only distal lymph nodes	0	0	2	6.9%	-
Other	2	6.9%	0	0	-
Two sites or more					
Liver and lymph nodes	3	10.3%	6	20.7%	0.4703
Liver and lung	1	3.4%	1	3.4%	-
Lung and lymph nodes	0	0	2	6.9%	-
Liver, lung and lymph nodes	1	3.4%	0	0	-
Liver, lung and other	1	3.4%	0	0	-

We further investigated the metastatic pattern of colorectal cancer based on a literature review. According to a retrospective study from the Surveillance, Epidemiology, and End Results Program (SEER) database based on 46,027 eligible patients with CRC, colon cancer had a higher incidence of liver metastasis than rectal cancer (13.8% vs. 12.3%), while rectal cancer had a higher incidence of lung (5.6% vs. 3.7%) metastasis than colon cancer.⁶³ A longitudinal retrospective analysis in 10,398 CRC patients after curative surgery reported that the incidence of liver and lung metastasis in patients with right-sided colon cancer, left-sided colon cancer, high rectal cancer and low rectal cancer were 2.66% and 1.01%, 4.45% and 1.69%, 5.29% and 4.28%, and 4.15% and 5.09%, respectively.⁶⁴ Another nationwide retrospective review of pathological records of 5817 CRC patients who underwent an autopsy showed that common adenocarcinoma predominantly metastasized to the liver compared with mucinous adenocarcinoma (73% vs. 52.2%) in both colon and rectal cancer.^{65,66} Moreover, a 20-year retrospective study of 2,286 CRC patients demonstrated that the incidence rate of synchronous lung metastasis was significantly lower than that of metachronous lung metastasis (2.8% vs. 10.2%). For metachronous lung metastasis, the incidence rates in colon cancer and rectal cancer were 6.6% and 15.1%, respectively. Collectively, in both colon cancer and rectal cancer, the liver was still the most common metastatic site, although low rectal cancer showed a slightly higher lung metastatic rate. Given that our patients in the DM group were all CRC patients with a common adenocarcinoma subtype and synchronous distant metastasis, it was not surprising that the proportion of pulmonary metastases was relatively low in the DM group.

References

63. Qiu M., et al. Pattern of distant metastases in colorectal cancer: a SEER based study. *Oncotarget* **6**, 38658-38666 (2015).
64. Augustad K. M., et al. Metastatic spread pattern after curative colorectal cancer surgery. A retrospective, longitudinal analysis. *Cancer Epidemiol* **39**, 734-744 (2015).
65. Hugen N., van de Velde C. J. H., de Wilt J. H. W. & Nagtegaal I. D. Metastatic pattern in colorectal cancer is strongly influenced by histological subtype. *Ann Oncol* **25**, 651-657 (2014).

66. Nozawa H., et al. Synchronous and metachronous lung metastases in patients with colorectal cancer: A 20-year monocentric experience. *Exp Ther Med* **3**, 449-456 (2012).

The reviewer's comment 3: line 284: The choice of 90% for specificity appears arbitrary. 95% would be more adequate in order to compare with other methods.

Authors' response: We thank the reviewer for the suggestion. We agree with the reviewer's opinion. According to the reviewer's suggestion, we chose 95% specificity to visualize the urinary protein performance for CRC diagnosis and metastatic risk prediction. We have revised Fig. 4e, Supplementary Fig. 3c, Supplementary Table S4d and 4e. The description in the Results section was also revised (Lines 285-294).

Revised Fig. 4e. Heatmap of the dot plot data for single urinary markers as well as the diagnostic or metastatic panel and the combination of corresponding clinical biomarker indices for the diagnostic or metastatic model in the training samples with a specificity of 95%. Red: positive using the cutoff value with a specificity of 95%. The FIT test, serum CEA, tumor location, sex and age are indicated by color coding. Neg., negative; Pos., positive; NA, not available.

Revised Supplementary Fig. 3c. Heatmap of the dot plot data for single urinary markers as well as the diagnostic or metastatic panel and the combination of corresponding clinical biomarker indices for the diagnostic or metastatic model in the validation samples with a specificity of 95%. The FIT test, serum CEA, tumor location, sex and age are indicated by color coding (right side). NA, not available.

The reviewer's comment 4: lines 294-305: Showing CORO1C tissue expression in metastasis would strengthen the association of this marker with the metastatic process.

Authors' response: We thank the reviewer for the suggestion. To clarify the relationship between CORO1C and metastasis at the tissue level, we performed an immunohistochemical staining assay in an additional eight multiple tissue microarrays (MTAs). Thus, a total of 961 tumors as well as 836 paracarcinoma normal tissues from 993 subjects with colorectal adenocarcinoma were informative for CORO1C. In the adjacent noncancerous tissues, the positivity rate of CORO1C staining was 5.6% (47/836), whereas it was 55.7% (535/961) in the CRC tumor tissues, showing significant upregulation in tumor tissues compared with the adjacent noncancerous tissues (revised Fig. 5a). Subsequent clinical significance analysis showed that higher CORO1C levels were associated with the depth of invasion (T staging, $P=0.0035$) and distant metastasis (M staging, $P=0.0210$; revised Fig. 5b). The Kaplan–Meier survival analysis indicated that CRC patients with high CORO1C protein levels had shorter recurrence-free survival (RFS) times ($P=0.0075$) but not overall survival times ($P=0.9171$; revised Fig. 6).

Revised Fig. 5 (partial). Immunohistochemical staining of CORO1C in tissues and its clinical significance. (a) Representative immunohistochemistry images and score distribution of CORO1C expression in paracarcinoma normal tissues (PN) and CRC tumors (tumor). The median and quartile values in each group of individuals are shown as thick red dotted line and thin purple dotted lines, respectively. Original magnification: $\times 200$. ****, $P < 0.0001$. (b) The

balloon plot for the clinical significance of CORO1C in colon adenocarcinoma patients with distinct staining intensities. The number in the circle is the sample size, and the percentage next to the circle is the corresponding percentage.

Revised Fig. 6 (partial). Kaplan–Meier survival analysis comparing overall survival (OS) and recurrence-free survival (RFS) of CORO1C in patients with colorectal adenocarcinoma. The numbers at the bottom of each panel indicate the patients at risk. Neg., negative expression; Pos., positive expression.

In addition, we analyzed an MTA including specimens from patients with distant metastases (n=6), negative lymph nodes (n=6) and positive lymph nodes (n=8). We also observed positive cytoplasmic staining for CORO1C in metastatic lesions in distant organs (the liver, lung and ovaries) and lymph nodes (Fig. H)

Figure H. Representative immunohistochemistry images of CORO1C in distant metastases and lymph nodes without metastasis (Neg LN) and with metastasis (Pos LN). Original magnification: upper panel, $\times 40$; bottom panel, $\times 200$.

Collectively, CORO1C tissue expression was indeed associated with the metastatic process..

The reviewer's comment 5: lines 307-317: The lack of effect of CORO1C overexpression or silencing on CRC cell growth, apoptosis or colony formation is surprising considering what has been reported in the literature with other cell models as well as the data shown with xenografts where proliferation is enhanced. This discrepancy should be addressed.

Authors' response: We thank the reviewer for the comment. Our results indicated that CORO1C had a very limited effect on *in vitro* cell proliferation, apoptosis or colony formation in DLD1, HCT116 and HCT8 cells. However, its overexpression or silencing significantly promoted *in vivo* cell growth and metastasis. The *in vitro* effect was inconsistent with the observations reported in gastric cancer, diffuse glioma and lung squamous cell carcinoma.⁶⁷⁻⁶⁹ We cannot exclude the influence of the tumor type or cell context. Moreover, these previous studies did not perform *in vivo* assays. However, our *in vivo* xenograft nude mouse model showed that CORO1C promotes cell growth.

Notably, the *in vitro* proliferation of cancer cells may not always reflect that *in vivo*.

A major drawback of *in vitro* assays is the failure to capture the inherent complexity of the tumor mass, organ systems, and the whole body. The tumor microenvironment is comprised of tumor cells, tumor stroma, blood vessels, a variety of resident and infiltrating host cells, secreted factors, and extracellular matrix proteins.^{70,71} Tumors cannot grow beyond 1–2 mm without a vascular supply due to an insufficient supply of nutrients and oxygen and poor clearance of metabolic waste, hypoxia, and/or acidosis.⁷² Thus, tumor progression is profoundly influenced by interactions of cancer cells with their environment that ultimately determine whether the primary tumor grows, is eradicated, or metastasizes. Thus, an *in vivo* animal model may account for the long-term cell proliferation, death, survival, and interactions between the cells and the host that support or restrain cell growth *in vivo*. For example, the overexpression of VEGFC in tumor cells does not influence *in vitro* proliferation but dramatically promotes *in vivo* growth in animal models.⁷³ In addition, CORO1C was identified as an extracellular stimulus and hypoxia-induced gene in fibroblasts, glioblastoma and head and neck squamous cell carcinoma cells.⁷⁴⁻⁷⁶ Thus, we hypothesized that xenografts in nude mice might induce Coro1c expression in stromal cells from host mice to support their proliferation.

More importantly, immune cells are the major and critical component of the tumor microenvironment. We evaluated the potential relationships between the expression of CORO1C and six types of tumor-infiltrating immune cells using TIMER online tool (<https://cistrome.shinyapps.io/timer/>). The results showed that CORO1C expression in colorectal cancer was positively correlated with the infiltrating level of CD8+ T cells ($\rho=0.33$, $P=8.96e-12$ for colon cancer; $\rho=0.359$, $P=1.42e-05$ for rectal cancer), macrophages ($\rho=0.419$, $P=1.25e-18$ for colon cancer; $\rho=0.195$, $P=2.12e-02$), neutrophils ($\rho=0.52$, $P=3.65e-29$ for colon cancer; $\rho=0.473$, $P=4.73e-09$ for rectal cancer), and dendritic cells ($\rho=0.467$, $P=3.26e-23$ for colon cancer; $\rho=0.305$, $P=2.61e-04$) (Fig. 1). It was reported that these intratumoral immune cells correlates with a malignant phenotype and poor prognosis in colorectal cancer.⁷⁷⁻⁸²

Figure I. Evaluation of CORO1C expression levels and immune infiltration levels in colorectal cancer with the TIMER tool based on TCGA RNA sequencing datasets. COAD, colon cancer; READ, rectal cancer..

Therefore, it is possible that CORO1C has no significant effect on cell proliferation *in vitro* experiments but facilitates the *in vivo* growth of CRC cells. However, according to the editor's comments, we removed the *in vitro* and *in vivo* functional assays on CORO1C to focus on the diagnostic/prognostic value of the urine protein signatures. Thus, these results and descriptions are not shown in our revised manuscript.

References

67. Cheng X., et al. CORO1C expression is associated with poor survival rates in gastric cancer and promotes metastasis *in vitro*. *FEBS Open Bio* **9**, 1097-1108 (2019).
68. Thal D., et al. Expression of coronin-3 (coronin-1C) in diffuse gliomas is related to malignancy. *J Pathol* **214**, 415-424 (2008).
69. Matak H., et al. Downregulation of the microRNA-1/133a cluster enhances cancer cell migration and invasion in lung-squamous cell carcinoma via regulation of Coronin1C. *J Hum Genet* **60**, 53-61 (2015).
70. Whiteside T. L. The tumor microenvironment and its role in promoting tumor growth. *Oncogene* **27**, 5904-5912 (2008).
71. De Palma M., Biziato D. & Petrova T.V. Microenvironmental regulation of tumour angiogenesis. *Nat Rev Cancer* **17**, 457-474 (2017).
72. Viallard C. & Larrivee B. Tumor angiogenesis and vascular normalization: alternative therapeutic targets. *Angiogenesis* **20**, 409-426 (2017).
73. Karpanen T., et al. Vascular endothelial growth factor C promotes tumor lymphangiogenesis and intralymphatic tumor growth. *Cancer Res* **61**, 1786-1790 (2001).
74. Chang H. Y., et al. Gene expression signature of fibroblast serum response predicts human cancer progression: similarities between tumors and wounds. *PLoS Biol* **2**, E7 (2004).
75. Tullai J. W., et al. Immediate-early and delayed primary response genes are distinct in function and genomic architecture. *J Biol Chem* **282**, 23981-23995 (2007).
76. Winter S. C., et al. Relation of a hypoxia metagene derived from head and neck cancer to

- prognosis of multiple cancers. *Cancer Res* **67**, 3441-3449 (2007).
77. Rao H. L., et al. Increased intratumoral neutrophil in colorectal carcinomas correlates closely with malignant phenotype and predicts patients' adverse prognosis. *PLoS One* **7**, e30806 (2012).
 78. Sandel M. H., et al. Prognostic value of tumor-infiltrating dendritic cells in colorectal cancer: role of maturation status and intratumoral localization. *Clin Cancer Res* **11**, 2576-2582 (2005).
 79. Malietzis G., et al. Prognostic Value of the Tumour-Infiltrating Dendritic Cells in Colorectal Cancer: A Systematic Review. *Cell Commun Adhes* **22**, 9-14 (2015).
 80. Yahaya M. A. F., et al. Tumour-Associated Macrophages (TAMs) in Colon Cancer and How to Reeducate Them. *J Immunol Res* 2368249 (2019).
 81. Kaplanski G., et al. IL-6: a regulator of the transition from neutrophil to monocyte recruitment during inflammation. *Trends Immunol* **24**, 25-9 (2003).
 82. Waldner M. J., Foersch S. & Neurath M. F. Interleukin-6 - A Key Regulator of Colorectal Cancer Development. *Int J Biol Sci* **8**, 1248-1253 (2012).

The reviewer's comment 6: lines 331-335/Figure 50: It is not clear that all of the proteins listed are "markedly increased" along with CORO-1C overexpression. Same for phosphorylation. For instance, both p-FAK and FAK appear to be increased in the same range so there is no net increase in phosphorylation, Same for c-Jun while there is a net decrease in SRC and JNK phosphorylation. Please revise.

Authors' response: We thank the reviewer for pointing out this inaccurate description. We agree with the reviewer's opinion that the expression of some total and phosphorylated forms of proteins were not "markedly" increased with CORO1C overexpression. The revised description: We observed that the expression of integrin α V, integrin β 1, FAK, SRC, ARHGEF7, RCC2, p70S6K, JNK and c-Jun but not RAC1, and 4EBP1 was markedly increased when CORO1C was overexpressed; and vice versa, CORO1C knockdown decreased the expression. Moreover, the activation of AKT and 4EBP1 was also altered along with the change in CORO1C protein expression, as measured by the ratio of the levels of the phosphorylated forms to the corresponding total protein levels. However, although it seems that there was a consistent trend for the levels of phosphorylated FAK, SRC, JNK and c-Jun after the forced expression and depletion of CORO1C, their phosphorylated/total ratio showed an inconsistent change.

However, according to the editor's comments, we removed the *in vitro* cell

function assays on CORO1C to focus on the diagnostic/prognostic value of the urine protein signatures. Thus, these new descriptions are not shown in our revised manuscript.

The reviewer's comment 7: lines 319-329: The basic xenograft model is useful for studying tumor cell growth at ectopic sites but other models addressing more specifically tumor invasion and metastasis in vivo would help to better define the involvement of CORO1C in CRC progression. For instance, direct injection of the cells into the mouse circulation via distinct sites (arterial, venous, portal vein, etc.) could be informative of the potential of organ colonization.

Authors' response:

[REDACTED]

We deeply appreciate your consideration of our manuscript. If you need additional information, please do not hesitate to contact us. We look forward to hearing from you.

With best regards,

Xiaohang Zhao, MD & Ph D, Professor, Cancer Hospital of CAMS

And

Wei Sun, Ph D, Associated Professor, Institute of Basic Medical Sciences of CAMS

REVIEWER COMMENTS

Reviewer #1 (Remarks to the Author):

The authors revised some parts well, according to my suggestion.

Responding to Comments from Reviewer 3:

The authors could carefully responded to most of comments well. However, the authors cannot solve the concerns of comment 6 and 15.

Comment 6: Information of Fig.1 is still insufficient for Independent Validation. As for diagnostic model, the numbers of Training set and Validation set are 171 and 86, respectively. The sum of them is not consistent with whole number of Validation stage. Similarly, the Metastatic model also raise same inconsistency. We are confused how the authors divided and selected the training and validation sets.

Comment 15: This is the most critical concern. In the Training set, the number of NM was only 27, but the authors analyzed 27 samples three times and counted those data as 81 NMs. Even if they analyzed 27 samples triplicated, the sample size is 27, not 81. This method is not scientifically appropriate. If this method is OK, anything is possible (ex. when sample size is only one, will 100 times-repeated analysis of a sample be $n=100?$). The authors must show the data from 27 samples.

Other comments:

1. The breakdown of Table 1 is unclear. A p-value for each factor should be analyzed among 4 groups, not among datasets of group. ie) Are there significant differences among 4 groups in the TMT, in the RPM and in the Dot blot??

In addition, age should be shown as the continuous value, and they should analyze statistical difference.

2. The authors described in the line 224/225: Moreover, RAD23B and GSPT2 concentrations were significantly higher in patients with LNM than in patients with NM ($p < 0.05$ for both).

However, no significant differences are noted in Fig. 4a. This description should be deleted or amended.

Reviewer #2 (Remarks to the Author):

All the problems have been solved.

Reviewer #4 (Remarks to the Author):

While the authors have clearly addressed all my previous remarks and concerns. We have been asked to comment also on the technical concerns raised by Reviewer #1 and, previously, by Reviewer #3.

We think all the concerns are related to the 'Independent validation' part in Figure 1. We agreed that oversampling is one of the techniques to avoid the class imbalance issue (author's reply to Reviewer #3's comment #15). However, there are two other issues raised by the other reviewers, which we also agree with.

First, how the data was split into training and validation is not clear (Reviewer #3's comment #6). Without clarification, a major concern is the bias introduced if the stratification is not rigorous. A

common solution to this is to perform a k-fold cross-validation or leave-one-out cross-validation. Such unbiased training/validation schemas have been adopted in close-related fields (Sun, 2020; Akbari, 2020; Rushing, 2015) and will make the result more convincing.

The other issue is the small sample size of the training data (concern on the validity of the performance given the limited number of validation samples). However, we think this issue could also be mitigated using leave-one-out cross-validation as every sample will be validated independently.

We went through the numbers, it seems the 434 samples should be a sum of the diagnostic model and the metastatic model. However, as mentioned by Reviewer #3 in comment #6, the number does not add up to 434 but 415.

Moreover, there is confusion on the terminology 'validation' in this section. In machine learning, model validation is referred to as the process where a trained model is evaluated with a testing data set. While per our understanding, the 'independent validation' in Figure 1 represents how the author built machine learning models to verify the validity of the selected biomarkers.

Reference:

Sun, Yinxiaohe, et al. "Epidemiological and clinical predictors of COVID-19." *Clinical Infectious Diseases* 71.15 (2020): 786-792.

Akbari, Hamed, et al. "Histopathology-validated machine learning radiographic biomarker for noninvasive discrimination between true progression and pseudo-progression in glioblastoma." *Cancer* 126.11 (2020): 2625-2636.

Rushing, Christel, et al. "A leave-one-out cross-validation SAS macro for the identification of markers associated with survival." *Computers in biology and medicine* 57 (2015): 123-129.

Reviewer #5 (Remarks to the Author):

I have no further comment. The authors have clearly addressed all my previous remarks and concerns.

I would like to express our sincere gratitude to the reviewers for their constructive and positive comments.

We have extensively revised the manuscript based on the reviewers' comments.

Below, please find our responses to the comments.

Reviewer #1 (Remarks to the Author):

The authors revised some parts well, according to my suggestion.

Authors' response: We sincerely appreciate all of the kindly comments.

Responding to Comments from Reviewer 3:

The authors could carefully responded to most of comments well. However, the authors cannot solve the concerns of comment 6 and 15.

Authors' response: We sincerely appreciate all of the positive comments.

Comment 6: Information of Fig.1 is still insufficient for Independent Validation. As for diagnostic model, the numbers of Training set and Validation set are 171 and 86, respectively. The sum of them is not consistent with whole number of Validation stage. Similarly, the Metastatic model also raise same inconsistency. We are confused how the authors divided and selected the training and validation sets.

Authors' response: We thank the reviewer for the comment. In the diagnostic and metastatic model development, to facilitate the comparison with commonly used clinical marker serum CEA, in the 434 urine samples detected by immunoassay, we excluded 122 samples without pre-operative CEA results (101 HCs and 21 CRC) from the subsequent modeling analysis. The rest 312 samples, including 154 HCs and 158 CRC patients (41 NM, 62 LNM and 55 DM), were used to verify the performance of our urinary protein diagnostic and metastatic signatures. In addition, according to the opinion of Reviewer #5, DM group might not be suitable for diagnostic model,

therefore, they were excluded in the diagnostic model. Thus, a total of 257 samples, consisted of 154 HCs, 41 CRC NM patients and 62 CRC LNM patients, were used in the diagnostic model. And a total of 158 CRC samples, consisted of 41 NM, 62 LNM and 55 DM patients, were used in the metastatic model.

For the 257 samples used in diagnosis model, 41 NM and 62 LNM patients were combined as the disease group indicating early- and intermediate-stage CRC. And 154 HC subjects were referred to as the control group. The block randomization method was used to split all samples into training set (~67% of dataset) and validation set (~33% of dataset). Thus, 171 subjects were enrolled in the training set (103 HC vs. 68 CRC) and 86 in the validation set (51 HCs vs. 35 CRC patients).

For the 158 samples used in metastatic model, 62 LNM and 55 DM patients were combined as the metastasis group indicating metastatic CRC, and 41 NM patients were assigned as nonmetastatic CRC. Initially, we also used block randomization method to divide these samples into training set (27 NM vs. 78 LNM+DM) and validation set (14 NM vs. 39 LNM+DM). Due to the small sample size of NM (27 samples) in training set, oversampling method was used in training set. According to your comments and the suggestions of Reviewer #4, we performed the leave-one-out cross-validation (LOOCV) instead of oversampling and random grouping methods.

The above description was revised in the Results section (in lines 227-232 and 244-245) and in the Materials and Methods section (in lines 645-647).

We also revised Fig. 1 according to your comments.

Revised Fig. 1. The overall workflow of study sample inclusion and exclusion criteria as well as the discovery, PRM verification, immunoassay verification and tissue validation for CRC urine biomarkers. The detailed inclusion and exclusion criteria of the samples are shown. CRC patients were divided into three groups by metastatic status: patients without metastases (NM), patients with lymph node metastasis (LNM) and patients with distant metastasis (DM). The four-stage workflow consisted of a series of mass spectrometry (MS) and immunoassay-based

approaches, including the tandem mass tag (TMT) labeling-2D-LC-MS/MS quantitative proteomic strategy, parallel reaction monitoring (PRM)-based targeted proteomic method, quantitative dot blot analysis and tissue immunohistochemistry (IHC), to construct a coherent and high-throughput cancer biomarker method in urine. CRC, colorectal cancer; MTA, multi-tissue array.

Comment 15: This is the most critical concern. In the Training set, the number of NM was only 27, but the authors analyzed 27 samples three times and counted those data as 81 NMs. Even if they analyzed 27 samples triplicated, the sample size is 27, not 81. This method is not scientifically appropriate. If this method is OK, anything is possible (ex. when sample size is only one, will 100 times-repeated analysis of a sample be $n=100$?). The authors must show the data from 27 samples.

Authors' response: We thank the reviewer for the comment. We agree with your opinion that 27 samples were triplicated to generate 81 data points but not 81 samples. Due to the small sample size of NM patients in training set (27 samples), oversampling method was used for the metastatic model. According to the suggestions of Reviewer #4, we used the leave-one-out cross-validation (LOOCV) method instead of oversampling and random grouping methods to construct metastatic model.

The results of LOOCV method showed that for the comparison of LNM+DM groups with NM group, the model yielded AUC, sensitivity and specificity of 0.699, 66.7%, 68.3% and 67.1%, respectively (Fig. 4c). And we found that the discriminative power of the metastatic signature was higher for DM versus NM (AUC 0.76) than for LNM versus NM (AUC 0.61) (Fig. S3b and Table S4c), highlighting its better performance for CRC with distant metastasis.

Fig. 4c. ROC curve of serum CEA, metastatic panel and the combination of the metastatic panel and CEA for metastatic model (LNM+DM vs NM).

Supplementary Fig. S3b. ROC curve of serum CEA, the metastatic panel and the combination of the metastatic panel and CEA for the metastatic model showing the discrimination between the LNM group or DM group and NM group.

Furthermore, we evaluated the complementarity of the urinary metastatic model for serum CEA measurements. For the ROC curves analyses, the overall performance of the urinary metastatic model was similar to serum CEA. And the combination of the urinary protein signature with serum CEA had a slightly better predictive power compared with CEA alone with AUC, sensitivity, specificity and accuracy values of 0.739, 70.9%, 73.2% and 71.5%, respectively (Fig. 4c and Table S4c). And in the stratified discrimination of LNM and DM from NM, the combination yielded AUCs of

0.659 and 0.831, respectively (Table S4c and Fig. S3b).

For the classification of metastatic and nonmetastatic CRC, at the most commonly used CEA threshold of 5 ng/mL in the clinic, the sensitivity, specificity and accuracy of serum CEA were 58.1%, 78.0% and 63.3%, respectively (Table S4c). In the patients with metastatic CRC, CEA was positive in 68 (58.1%) patients, and the urine metastatic signature increased the diagnostic power in an additional 29 (24.8%) patients. The subgroup analyses revealed that the discriminative power of the metastatic model increased in an additional 17 (27.4%) and 12 (21.8%) patients with LNM and DM, respectively, compared with that of CEA alone in 29 (46.8%) and 39 (70.9%) patients (Fig. 4d). Moreover, 51.5% (17 out of 33) or 75.0% (12 out of 16) of the CEA-negative LNM or DM patients were correctly predicted to have metastases using the panel.

Fig. 4d. Metastatic predictive power of the metastatic signature in the individuals who were misdiagnosed by the serum CEA. The values in parentheses indicate the number of samples corresponding to each percent.

Above results from leave-one-out cross-validation method were similar to the results from oversampling method. We updated the description in the Results section

(in lines 244-250, 266-281 and 288) and in the Materials and Methods section (in lines 645-647 and 690-691) of our revised manuscript. Meanwhile, the Fig. 4, Fig. S3 and Table S4 were also modified.

Other comments:

1. The breakdown of Table 1 is unclear. A p-value for each factor should be analyzed among 4 groups, not among datasets of group. ie) Are there significant differences among 4 groups in the TMT, in the RPM and in the Dot blot??

In addition, age should be shown as the continuous value, and they should analyze statistical difference.

Authors' response: We thank the reviewer for the comment and suggestion. Table 1 was reorganized and the *P* values for each factor among four groups were analyzed. Meanwhile, age was shown as the continuous variable as the format of “median (minimum - maximum)”. The updated Table 1 was shown as below and in our revised manuscript.

Additionally, the description of Table 1 was also revised in lines 112-118 in the Results section: “The age and sex distributions were balanced among the HCs and three groups of CRC patients, except that the samples of HC group in the dot blot analysis were a little younger. In addition, there were no statistically significant differences in the clinical parameters of histological differentiation grade, CA19-9 level and tumor location among the three CRC groups (NM, LMN and DM) in the TMT and PRM analysis; CEA level, CA19-9 level and tumor location showed significant difference among the three CRC groups in the dot blot analysis.”

Table 1. Clinical information of all samples used in this study.

	TMT				P value ^a	PRM				P value	Dot blot				P value
	HC	NM	LNM	DM		HC	NM	LNM	DM		HC	NM	LNM	DM	
Sex					0.6482					0.3792					0.1811
Male	4	6	3	4		14	14	10	18		153	34	51	40	
Female	5	3	6	5		11	7	3	5		102	12	24	18	
Age (years)					0.6354 ^b					0.3971 ^b					<0.0001 ^b
median (min-max)	48 (42-66)	56 (35-71)	57 (34-75)	50 (40-80)		55 (40-68)	56 (31-74)	56 (45-80)	58 (40-68)		56 (23-78)	61 (26-78)	62 (31-87)	63 (39-87)	
Histological grade					0.0513					0.9537 ^a					0.4372
Well differentiated		2	0	1			1	0	0			1	0	1	
Moderately differentiated		7	5	5			16	9	16			30	40	31	
Poorly differentiated		0	4	0			4	1	3			14	30	18	
Unknown		0	0	3			0	3	4			1	5	8	
CEA (ng/mL)					0.0491					0.0044 ^a					<0.0001 ^c
<5	-	4	3	2		-	6	4	5		147	32	34	16	
≥5	-	1	0	6		-	0	2	13		7	9	32	39	
Unknown	-	4	6	1		-	15	7	5		101	5	9	3	
CA19-9 (U/mL)					0.0504					0.7809 ^a					<0.0001 ^c
<37	-	5	3	3		-	5	6	11		152	40	56	35	
≥37	-	0	0	5		-	1	0	3		2	1	7	20	
Unknown	-	4	6	1		-	15	7	9		101	5	12	3	
Tumor location					0.2977					0.1463 ^a					0.0083
Right sided		2	1	2			4	2	4			6	8	11	
Left sided		5	2	5			3	3	11			20	13	18	
Rectum		2	6	2			14	8	8			20	54	29	
TNM staging					<0.0001					<0.0001					<0.0001
I		0	0	0			11	0	0			11	0	0	
II		9	0	0			10	0	0			35	0	0	
III		0	9	0			0	13	0			0	75	0	
IV		0	0	9			0	0	23			0	0	58	

^a, Fisher's Exact Test; ^b, One way ANOVA or Kruskal-Wallis One Way Analysis of Variance; ^c, statistical analysis only among three CRC disease groups

2. The authors described in the line 224/225: Moreover, RAD23B and GSPT2 concentrations were significantly higher in patients with LNM than in patients with NM ($p < 0.05$ for both). However, no significant differences are noted in Fig. 4a. This description should be deleted or amended.

Authors' response: We thank the reviewer for pointing out this issue. We have deleted this description in our revised manuscript.

Reviewer #2 (Remarks to the Author):

All the problems have been solved.

Authors' response: We sincerely appreciate all of the kindly comments.

Reviewer #4 (Remarks to the Author):

While the authors have clearly addressed all my previous remarks and concerns. We have been asked to comment also on the technical concerns raised by Reviewer #1 and, previously, by Reviewer #3.

Authors' response: We sincerely appreciate all of the kindly comments.

We think all the concerns are related to the 'Independent validation' part in Figure 1. We agreed that oversampling is one of the techniques to avoid the class imbalance issue (author's reply to Reviewer #3's comment #15). However, there are two other issues raised by the other reviewers, which we also agree with.

First, how the data was split into training and validation is not clear (Reviewer #3's comment #6). Without clarification, a major concern is the bias introduced if the stratification is not rigorous. A common solution to this is to perform a k-fold cross-validation or leave-one-out cross-validation. Such unbiased training/validation schemas have been adopted in close-related fields (Sun, 2020; Akbari, 2020; Rushing, 2015) and will make the result more convincing.

The other issue is the small sample size of the training data (concern on the validity of the performance given the limited number of validation samples). However, we think this issue could also be mitigated using leave-one-out cross-validation as every

sample will be validated independently.

Authors' response: We thank the reviewer for the kind and constructive suggestion. For the construction of metastatic model, we previously used block randomization method to divide 158 samples into training set (27 NM vs. 78 LNM+DM) and validation set (14 NM vs. 39 LNM+DM). According to your suggestion, we performed the leave-one-out cross-validation (LOOCV) analysis in metastatic model. The data files and source code had been uploaded to the folder of “metastatic model for dot plot data” in the GitHub (<https://github.com/scshaochen/PRMColonCancer/tree/main/metastatic%20model%20for%20dot%20plot%20data>).

The results from leave-one-out cross-validation method showed that for the discriminative of LNM+DM groups and NM group, the model yielded AUC, sensitivity and specificity of 0.699, 66.7%, 68.3% and 67.1%, respectively (Fig. 4c). And the discriminative power of the metastatic signature was higher for DM versus NM (AUC 0.76) than for LNM versus NM (AUC 0.61) (Fig. S3b and Table S4c), highlighting its better performance for CRC with distant metastasis.

Fig. 4c. ROC curve of serum CEA, metastatic panel and the combination of the metastatic panel and CEA for metastatic model (LNM+DM vs NM).

Supplementary Fig. S3b. ROC curve of serum CEA, the metastatic panel and the combination of the metastatic panel and CEA for the metastatic model showing the discrimination between the LNM group or DM group and NM group.

Furthermore, we evaluated the complementarity of the urinary metastatic model for serum CEA measurements. For the ROC curves analyses, the overall performance of the urinary metastatic model was similar to serum CEA. And the combination of the urinary protein signature with serum CEA had a slightly better prediction compare with CEA alone, with AUC, sensitivity, specificity and accuracy values of 0.739, 70.9%, 73.2% and 71.5%, respectively (Fig. 4c and Table S4c). And in the stratified discrimination of LNM and DM from NM, the combination yielded AUCs of 0.659 and 0.831, respectively (Table S4c and Fig. S3b).

For the classification of metastatic and nonmetastatic CRC, at the most commonly used CEA threshold of 5 ng/mL in the clinic, the sensitivity, specificity and accuracy of serum CEA were 58.1%, 78.0% and 63.3%, respectively (Table S4c). In the patients with metastatic CRC, CEA was positive in 68 (58.1%) patients, and the metastatic signature increased the diagnostic power in an additional 29 (24.8%) patients. The subgroup analyses revealed that the diagnostic power of the metastatic model increased in an additional 17 (27.4%) and 12 (21.8%) patients with LNM and DM, respectively, compared with that of CEA alone in 29 (46.8%) and 39 (70.9%) patients (Fig. 4d). Moreover, 51.5% (17 out of 33) or 75.0% (12 out of 16) of the CEA-negative LNM or DM patients were correctly predicted to have metastases using the panel.

Fig. 4d. Metastatic predictive power of the metastatic signature in the individuals who were misdiagnosed by the serum CEA. The values in parentheses indicate the number of samples corresponding to each percent.

The Above results from leave-one-out cross-validation method were similar to the results of oversampling method. We noticed that the AUC value under the ROC curve (0.699) from the leave-one-out method was lower than the previous AUC value in the training set (0.77) from oversampling method, but similar to that in the validation set (0.68). We thought it may be due to the small sample size of NM group, thus, overfitting and grouping bias might be introduced. Therefore, we used the results of leave-one-out cross-validation method in the revised manuscript.

We updated the description in the Results section (in lines 244-250, 266-281 and 288) and in the Materials and Methods section (in lines 6452-647 and 690-691) of our revised manuscript. Meanwhile, the Fig. 4, Fig. S3 and Table S4 were also modified.

We greatly appreciate the reviewer's nice advice. Thanks again for helping us improve the results.

We went through the numbers, it seems the 434 samples should be a sum of the

diagnostic model and the metastatic model. However, as mentioned by Reviewer #3 in comment #6, the number does not add up to 434 but 415.

Authors' response: We thank the reviewer for the comment. In the diagnostic and metastatic model development, to facilitate the comparison with commonly used clinical marker serum CEA, in the 434 urine samples detected by immunoassay, we excluded 122 samples without pre-operative CEA results (101 HCs and 21 CRC) from the subsequent modeling analysis. The rest 312 samples, including 154 HCs and 158 CRC patients (41 NM, 62 LNM and 55 DM), were used to verify the performance of our urinary protein diagnostic and metastatic signatures. In addition, according to the opinion of Reviewer #5, DM group might not be suitable for diagnostic model, therefore, they were excluded in the diagnostic model. Thus, a total of 257 samples, consisted of 154 HCs, 41 CRC NM patients and 62 CRC LNM patients, were used in the diagnostic model. And a total of 158 CRC samples, consisted of 41 NM, 62 LNM and 55 DM patients, were used in the metastatic model.

Taken together, there are overlapping samples used in the diagnostic model and metastatic model. The above description was seen in the Results (in lines 227-230 and 244-245) and Materials and Methods (in lines 641-647 and 688-692) sections.

We also revised Fig. 1 according to your comments.

Revised Fig. 1. The overall workflow of study sample inclusion and exclusion criteria as well as the discovery, PRM verification, immunoassay verification and tissue validation for CRC urine biomarkers. The detailed inclusion and exclusion criteria of the samples are shown. CRC patients were divided into three groups by metastatic status: patients without metastases (NM), patients with lymph node metastasis (LNM) and patients with distant metastasis (DM). The four-stage workflow consisted of a series of mass spectrometry (MS) and immunoassay-based

approaches, including the tandem mass tag (TMT) labeling-2D-LC-MS/MS quantitative proteomic strategy, parallel reaction monitoring (PRM)-based targeted proteomic method, quantitative dot blot analysis and tissue immunohistochemistry (IHC), to construct a coherent and high-throughput cancer biomarker method in urine. CRC, colorectal cancer; MTA, multi-tissue array.

Moreover, there is confusion on the terminology 'validation' in this section. In machine learning, model validation is referred to as the process where a trained model is evaluated with a testing data set. While per our understanding, the 'independent validation' in Figure 1 represents how the author built machine learning models to verify the validity of the selected biomarkers.

Authors' response: We thank the reviewer for the comment. We completely agree with the reviewer's opinion. We replaced the words "Independent Validation" with "Immunoassay Verification" in Fig. 1 as shown as above. Meanwhile, the relevant text in the Results section (in lines 214-215) was also modified in our revised manuscript.

Reference:

*Sun, Yinxiaohe, et al. "Epidemiological and clinical predictors of COVID-19." *Clinical Infectious Diseases* 71.15 (2020): 786-792.*

*Akbari, Hamed, et al. "Histopathology - validated machine learning radiographic biomarker for noninvasive discrimination between true progression and pseudo - progression in glioblastoma." *Cancer* 126.11 (2020): 2625-2636.*

*Rushing, Christel, et al. "A leave-one-out cross-validation SAS macro for the identification of markers associated with survival." *Computers in biology and medicine* 57 (2015): 123-129.*

Authors' response: We thank the reviewer for providing these helpful references about leave-one-out cross-validation.

Reviewer #5 (Remarks to the Author):

I have no further comment. The authors have clearly addressed all my previous remarks and concerns.

Authors' response: We sincerely appreciate all of the kindly comments.

Finally, we revised figures (Figs. 1, 4 and S3), tables (Tables 1 and S4) and the manuscript according to the reviewers' comments. We believe that we have addressed all concerns raised by the reviewers. We hope that you will find our revised manuscript acceptable.

REVIEWER COMMENTS

Reviewer #1 (Remarks to the Author):

The manuscript has been appropriately revised.

Reviewer #4 (Remarks to the Author):

All issues are addressed.

Reviewer #1 (Remarks to the Author):

The manuscript has been appropriately revised.

Authors' response: We sincerely appreciate all of the kindly comments.

Reviewer #4 (Remarks to the Author):

All issues are addressed.

Authors' response: We sincerely appreciate all of the kindly comments